# Inflammation-induced IgE promotes epithelial hyperplasia and tumour growth

Mark David Hayes[1], Sophie Ward[1], Greg Crawford[1], Rocio Castro Seoane[1], William David Jackson[1], David Kipling[2], David Voehringer[3], Deborah Dunn-Walters[4], Jessica Strid[1]*

[1]Department of Immunology and Inflammation, Imperial College London, London, United Kingdom; [2]Division of Cancer and Genetics, School of Medicine, Cardiff University, Cardiff, United Kingdom; [3]Department of Infection Biology, University Hospital Erlangen and Friedrich-Alexander University Erlangen-Nuremberg (FAU), Erlangen, Germany; [4]Faculty of Health and Medical Sciences, School of Biosciences and Medicine, University of Surrey, Guildford, United Kingdom

**Abstract** IgE is the least abundant circulating antibody class but is constitutively present in healthy tissues bound to resident cells via its high-affinity receptor, FcεRI. The physiological role of endogenous IgE antibodies is unclear but it has been suggested that they provide host protection against a variety of noxious environmental substances and parasitic infections at epithelial barrier surfaces. Here we show, in mice, that skin inflammation enhances levels of IgE antibodies that have natural specificities and a repertoire, VDJ rearrangements and CDRH3 characteristics similar to those of IgE antibodies in healthy tissue. IgE-bearing basophils are recruited to inflamed skin via CXCL12 and thymic stromal lymphopoietin (TSLP)/IL-3-dependent upregulation of CXCR4. In the inflamed skin, IgE/FcεRI-signalling in basophils promotes epithelial cell growth and differentiation, partly through histamine engagement of $H_1R$ and $H_4R$. Furthermore, this IgE response strongly drives tumour outgrowth of epithelial cells harbouring oncogenic mutation. These findings indicate that natural IgE antibodies support skin barrier defences, but that during chronic tissue inflammation this role may be subverted to promote tumour growth.

*For correspondence:
j.strid@imperial.ac.uk

Competing interests: The authors declare that no competing interests exist.

## Introduction

IgE is present at low concentrations in the blood and is the least abundant circulating antibody class. However, all epithelial tissues contain resident cells constitutively binding IgE antibodies. Although the half-life of free IgE in the serum is short (1–2 days), tissue-resident IgE may persists for weeks or months (*Geha et al., 2003*). In the tissue, IgE is bound with high affinity to its receptor, FcεRI, and can remain bound for the life of the FcεRI-bearing cell (*MacGlashan, 2008*). The specificity and function of this endogenous IgE in healthy individuals is unclear, although spontaneous production of natural IgE has been suggested to be a component of innate tissue defence (*McCoy et al., 2006*; *Palm et al., 2012*).

Mast cells and basophils are the only two cell types that express the complete high-affinity receptor for IgE (FcεRI) with all four polypeptide subunits ($\alpha\beta\gamma_2$). In humans, subsets of dendritic cells, monocytes and eosinophils can express low levels of FcεRI but only as a trimer, lacking the β chain ($\alpha\gamma_2$). The density of mast cell and basophil FcεR1 expression correlates with serum IgE levels, so an increase in IgE antibody may directly increase FcεRI signalling (*Lawrence et al., 2017*). Mast cells and basophils are potent innate effector cells that have overlapping effector functions. Paul Ehrlich first described mast cells and basophils in 1878, and estimated that there were enough mast cells in the body to make up 'an organ the size of the spleen' (*Crivellato et al., 2003*). Mast cells are constitutively resident in tissues, such as the skin, whereas basophils are circulating cells that rapidly

infiltrate inflamed tissues. Crosslinking of FcεRI-bound IgE on these cells causes degranulation and immediate release of potent pharmacologically active pre-formed mediators and concurrent synthesis of cytokines/chemokines and inflammatory lipid mediators. Although the in vivo role(s) of both cell types is only partially understood, it has been suggested that they mediate a 'gate-keeper' function and promote barrier defences at the skin and mucosal surfaces (*Voehringer, 2013*).

IgE antibodies have been studied most commonly in the context of atopic allergic diseases or parasitic infections. However, it has been proposed that they also play an important protective role in host defence against noxious environmental substances such as venoms, hematophagous fluids, environmental xenobiotics and irritants (*Palm et al., 2012*; *Profet, 1991*). In support of this, we recently demonstrated that topical exposure to environmental carcinogens promotes potent de novo induction of autoreactive IgE antibodies, and that these IgE antibodies provide protection against epithelial carcinogenesis in the exposed tissue (*Crawford et al., 2018*). The induction of IgE was dependent on epithelial cell (EC) DNA-damage, and deep-sequencing of these IgE antibodies revealed that they have a unique repertoire with specific VDJ rearrangements and CDRH3 characteristics that are distinct from those of IgE produced during general tissue inflammation (*Crawford et al., 2018*).

Epidermal hyperplasia and inflammation are hallmarks of a wide range of skin conditions. Furthermore, chronic inflammation may aid the proliferation and survival of EC harbouring genomic alterations, and so is now widely accepted to promote cancer development (*Mantovani et al., 2008*; *Coussens et al., 2013*; *Grivennikov et al., 2010*). Here, we explored the nature of inflammation-induced IgE antibodies and their role in tissue homeostasis. We show that general tissue inflammation enhances the levels of polyclonal natural IgE with similar characteristics and VDJ usage as IgE in naïve animals. Furthermore, this IgE, via FcεRI-signalling in basophils, potently promotes epidermal hyperplasia and inflammation-driven outgrowth of skin tumours following subclinical carcinogen exposure. Tissue inflammation recruits IgE-bearing basophils to the skin via CXCL12 and TSLP/IL-3-dependent upregulation of CXCR4 on circulating basophils. IgE-dependent basophil degranulation in the skin supports EC proliferation and differentiation, partly via histamine. Overall, our studies demonstrate how endogenous IgE antibodies that have natural specificities potently affect skin EC growth and differentiation with implications for both atopy and cancer.

## Results

### Skin inflammation increases IgE levels locally and systemically

Both basophils and mast cells carry a high number of FcεRI receptors, which are usually occupied with IgE (*Lawrence et al., 2017*). Consistent with this, a horizontal view of resting healthy murine skin revealed that large amounts of IgE was normally present in the tissue (*Figure 1a*) despite very low levels in serum (*Figure 1b–d*). IgE in resting skin was mainly found on cells around the hair-follicles (*Figure 1a*). Topical exposure to agents that induce skin inflammation, such as 12-*O*-tetradecanoylphorbol-13-acetate (TPA, a protein kinase C activator) (*Figure 1b*), MC903 (a vitamin D3 analogue commonly used to induce atopic dermatitis-like inflammation) (*Figure 1c*) and R848 (resiquimod, a toll-like receptor seven agonist commonly used to induce psoriasis-like inflammation) (*Figure 1d*), significantly enhanced the circulating levels of IgE compared to those in untreated or vehicle-treated animals. Enhancement of serum IgE was dependent on topical exposure: intravenous (i.v.) or intraperitoneal (i.p.) administration did not trigger the same effect, as shown by R848 exposure (*Figure 1d*).

The increase in serum IgE was accompanied by an increase in the number of IgE-secreting plasma cells in the skin-draining lymph nodes (LNs) (*Figure 1e,f*). IgE levels were also increased locally in the skin following topical TPA treatment (*Figure 1g*), but most notably the cells carrying the IgE switched from being predominantly mast cells in resting untreated skin to mainly basophils in TPA-treated inflamed skin (*Figure 1h,i*). 24 hr after cessation of TPA treatment, only a few mast cells remained in the skin, while IgE-bearing basophils accounted for ~2% of total CD45$^+$ leukocytes. Basophil numbers further increased at 48 hr after cessation of TPA treatment and then declined as the inflammation subsided, while mast cells returned to the skin (*Figure 1i*). Thus, untreated resting skin contains high levels of IgE antibodies that are predominately carried on mast cells. Skin

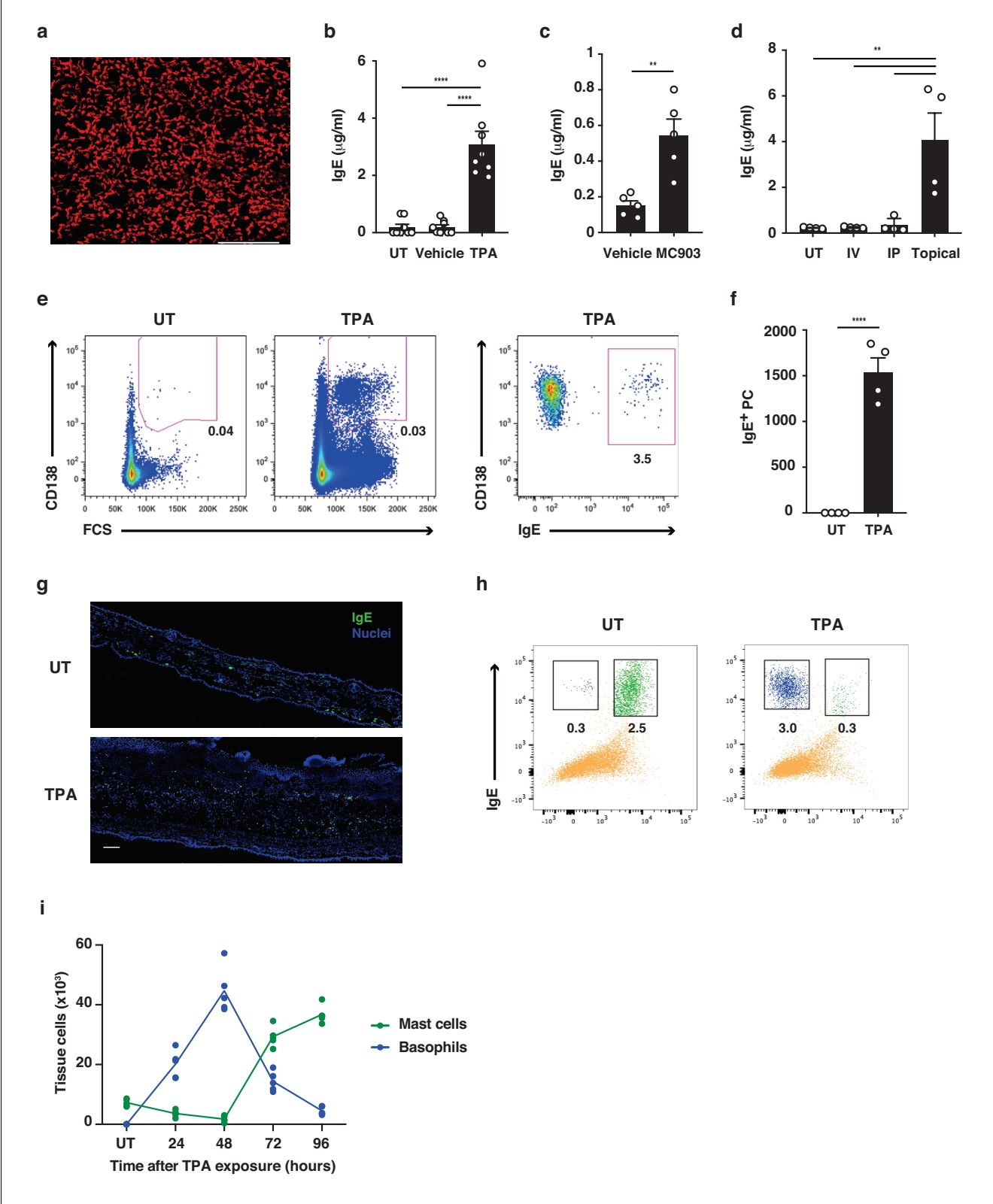

**Figure 1.** Skin inflammation increases IgE levels locally and systemically. (a) Representative image of IgE staining (red) in healthy dermal skin. Scale = 50 μm. The image is representative of tile-scans from healthy untreated (UT) dermal sheets of five independent mice. (b–d) ELISA of IgE in serum of UT wildtype (WT) BALB/c mice and mice treated on the dorsal ear skin with (b) 2.5 nM TPA 2x a week for 2 weeks or vehicle control (ethanol) (n = 8), (c) 1 nM MC903 5x a week for 2 weeks or vehicle control (ethanol) (n = 5) or (d) 100 μg R848 3x a week for 2 weeks or similarly i.v. or i.p. (n = 4).

*Figure 1 continued on next page*

*Figure 1 continued*

Data are expressed as means ± SEM. (**e, f**) Fluorescence-assisted cell sorting (FACS) analysis of IgE-secreting plasma cells in the skin draining LNs of UT WT mice and mice exposed topically to TPA on the dorsal side of the ears 2x a week for 2 weeks. (**e**) Representative plots of plasma cells gated as FSC$^{hi}$CD95$^+$CD138$^+$ cells, and with intracellular IgE staining to show isotype switching. (**f**) Enumeration of IgE$^+$ plasma cells in skin draining LN (n = 4). (**g**) Representative images of IgE staining (green) in UT and TPA treated whole skin. Nuclei in blue. Scale = 100 μm. Images are representative of tile-scans from ear cross-sections from six independent mice. (**h, i**) FACS analysis of IgE-bearing cells in whole naïve UT skin and in TPA-treated skin (TPA 2x a week for 2 weeks). (**h**) Representative plots of IgE-bearing mast cells (green) and basophils (blue) 48 hr after last TPA exposure. (**i**) Enumeration of IgE-bearing mast cells and basophils at the indicated time points after topical TPA. Mast cells were defined as CD45$^{hi}$cKit$^+$IgE$^+$CD41$^-$ and basophils as CD45$^{lo}$cKit$^-$IgE$^+$CD41$^+$(n = 5). One-way ANOVA multiple comparison (**b, d**) and two-tailed Student's t-test for unpaired data (**c, f**) were used to test for statistical difference. **p<0.01 and ****p<0.0001. IP, intraperitoneal; IV, intravenous; UT, untreated.

inflammation increases IgE levels locally and systemically, and IgE antibodies in inflamed skin are mainly carried on basophils.

## Skin inflammation enhances polyclonal 'natural IgE' with characteristics similar to those in naïve animals

To understand the nature of IgE-induced during skin inflammation, we performed high-throughput sequencing of the IgE heavy-chain repertoire from mice exposed topically to TPA. This revealed the repertoires to be highly diverse and polyclonal with ~75% of the repertoire consisting of clone sizes ≤ 10 (*Figure 2a*), which was similar to that seen in untreated naïve mice. This suggests that no notable selection or clonal expansion had occurred. The most prominent effects of B cell selection can be seen in heavy-chain gene-usage and in CDRH3 characteristics. Thus, we analysed the use of genes encoding the variable, diversity and joining (VDJ) regions of the IgE heavy-chain and found that IgHV1 was the most dominant Igh-V family gene, accounting for >70% of the repertoire (*Figure 2b*). All four Igh-J family genes and six Igh-D family genes were used, although IgHD2 dominated (*Figure 2c*). The overall VDJ family gene usage in untreated naïve mice and in the TPA-treated mice was similar (*Figure 2b,c*). Further, we analysed the characteristics and physicochemical properties of the complementarity-determining CDRH3 regions because these form an important part of the antigen-binding site of the antibody. We found no differences in CDRH3 length, levels of mutation, aliphatic index or isoelectric point (pI) between the IgE sequenced from TPA-treated or untreated mice (*Figure 2d*), suggesting that the TPA-induced skin inflammation did not alter the composition or the nature of the IgE response. The dominant use of the V1 family and the low mutation rate are consistent with those of 'natural IgE' (*McCoy et al., 2006*), thus we next tested whether the inflammation-induced IgE repertoire contained the characteristic natural anti-phosphorylcholine (anti-PC) (*Kearney, 2005*) specificity. This analysis showed that skin inflammation enhanced the production of anti-PC IgE to a level similar to that of the spontaneous production of natural IgE seen in immunodeficient *Tcrb*$^{-/-}$ mice (*Figure 2e*). Levels of circulating anti-PC IgE increased further in skin-tumour-bearing mice that had been exposed to TPA for 20 weeks (*Figure 2e*). Together, this analysis revealed that skin inflammation enhances levels of undiversified natural IgE akin to the IgE spontaneously present at low levels in healthy mice.

## IgE promotes inflammation-driven outgrowth of skin tumours

We have recently reported that IgE antibodies that are induced de novo following repeated topical exposure to environmental carcinogens is protective in a mutation-driven cutaneous carcinogenesis model, and that the repertoire of the carcinogen-induced IgE differed substantially from the IgE induced by general skin inflammation (*Crawford et al., 2018*). We therefore tested whether natural IgE would influence cutaneous carcinogenesis in an inflammation (TPA)-driven model. This question was explored in a widely used two-stage model, in which topical exposure to a subclinal dose of the carcinogen 7,12-dimethylbenz[a]anthracene (DMBA) provokes few oncogenic mutations in EC (not enough for cancer growth) that can subsequently be promoted to grow into overt tumours by chronic tissue inflammation (*Abel et al., 2009*). Mice lacking IgE (*Igh7*$^{-/-}$) were markedly protected from developing tumours in this two-stage model (*Figure 3a*). Although *Igh7*$^{-/-}$ mice showed an equivalent level of DNA-damage to WT following carcinogen exposure (*Figure 3—figure supplement 1a*), as assessed by staining for the phosphorylated histone H2A variant H2AX (γH2AX), only a few *Igh7*$^{-/-}$ mice developed tumours and the ones that did grew only very few and

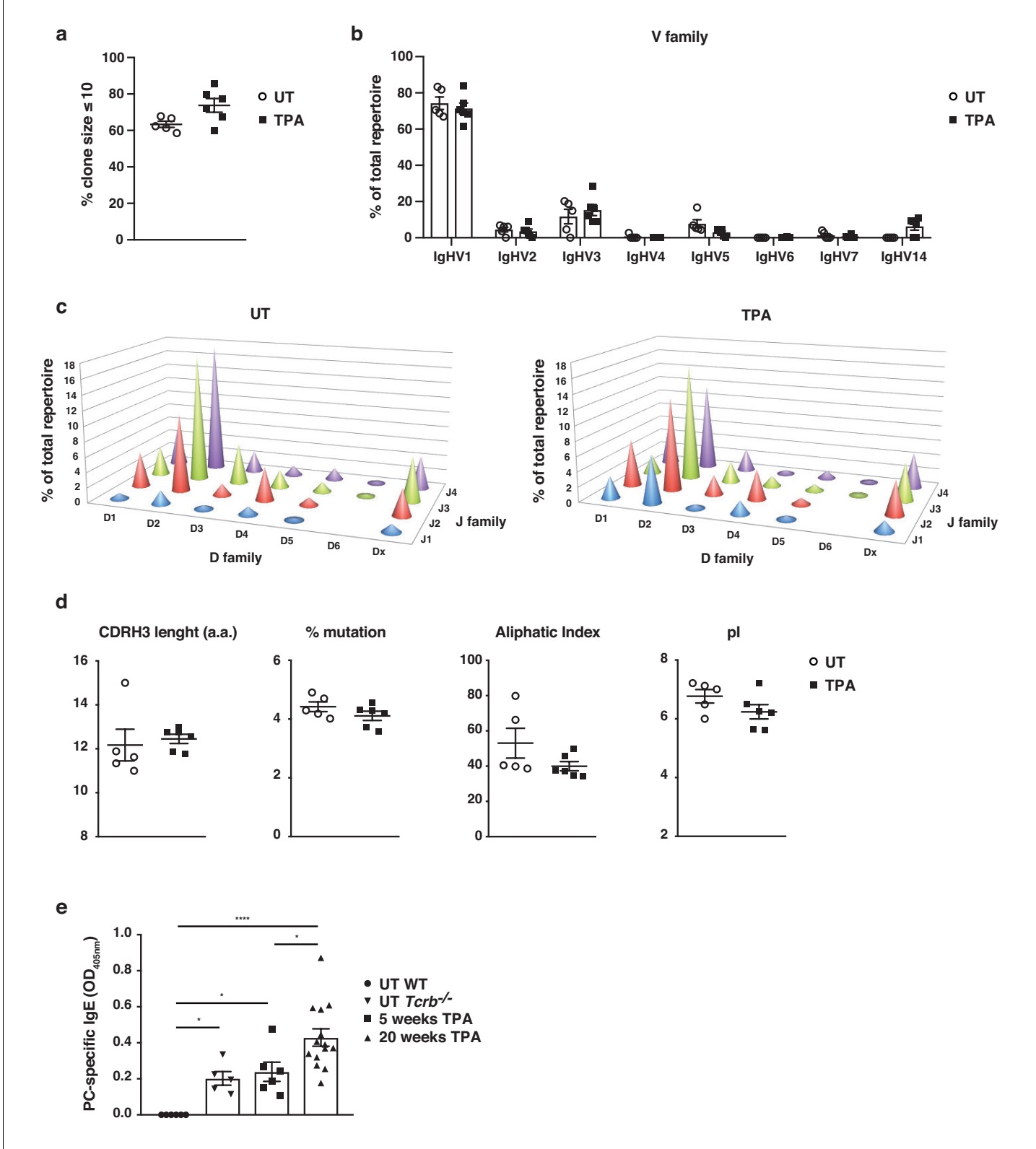

**Figure 2.** Skin inflammation induces polyclonal 'natural IgE' antibodies with characteristics similar to those of IgE antibodies in naïve animals. (a–d) High-throughput sequencing and heavy-chain repertoire analysis of IgE in sorted $FSC^{hi}CD95^{hi}CD138^+$ plasma cells from skin-draining LNs of wildtype (WT) mice 7 days after topical exposures to TPA (TPA 2x a week for 2 weeks) (n = 6) and in whole spleen from untreated (UT) naïve mice (n = 5). (a) % of the total repertoire consisting of clone sizes ≤ 10. (b) Igh-V gene family usage expressed as mean of total repertoire ± SEM. V genes
*Figure 2 continued on next page*

*Figure 2 continued*

that are not used in the repertoire of any mouse are not included for clarity. (**b**) Paired DJ repertoire shown as average Igh-D and Igh-J use of total repertoire in UT (left) and TPA-treated (right) mice. (**d**) Physicochemical properties of the CDRH3 regions; length (number of amino acids), % mutation compared to germline, aliphatic index and isoelectric point (pI) presented as mean ± SEM. (**e**) Level of IgE in serum specific for phosphorylcholine (PC) in UT WT (n = 6), UT *Tcrb*$^{-/-}$ (n = 5), 10x TPA-treated (2x/week) WT mice (n = 6) and in WT mice that have undergone 7,12-dimethylbenz[a] anthracene (DMBA)-TPA carcinogenesis (single low-dose DMBA + 20 weeks TPA, 2x/week) (n = 14). Data are presented as means ± SEM. Statistics by two-tailed Student's t-test for unpaired data (**a, b, d**) and one-way ANOVA multiple comparison (**e**); *p<0.05 and ****p<0.0001.

significantly smaller papillomas (*Figure 3a*), suggesting that the lack of IgE during TPA promotion hindered tumour growth. The topical TPA application enhanced the circulating levels of IgE in WT animals throughout the experiment (*Figure 3b*). *Igh7*$^{-/-}$ mice developed no IgE and reduced levels of IgG1 and IgG2a compared to WT animals (*Figure 3—figure supplement 1b*). Analysis of the tumour tissue and the peri-lesional skin by flow cytometry (*Figure 3c*) and by qRT-PCR (*Figure 3d*) showed that basophils were the predominant cells carrying IgE in the tumours with very few mast cells entering the tumour. The peri-lesional skin contained both mast cells and basophils, whereas mast cells dominated in untreated belly skin from the same animals (*Figure 3c,d*). Cross-sections of whole tumours showed that IgE-bearing cells accumulated right up to the tumour, mainly in the peri-tumoural infiltrate (*Figure 3e*), with some also entering the epithelium (*Figure 3e* inset). Thus, TPA-enhanced accumulation of IgE-bearing cells potently promotes the outgrowth of inflammation-driven tumours, with IgE-bearing basophils accumulating inside skin tumours.

## IgE exacerbates skin inflammation by altering the microenvironment and inducing epithelial hyperplasia

To further explore the role of IgE antibodies during TPA-induced inflammation (the tumour promotion phase), we investigated the skin microenvironment following repeated topical application of TPA on its own to the ear skin. This showed that TPA caused substantial de novo infiltration of CD45$^+$ leukocytes into the skin, which was equal in WT and *Igh7*$^{-/-}$ mice (*Figure 4a*). There were no substantial differences in the cellular composition of the infiltrates, aside from a minor increase in dermal Vγ5$^-$ γδ T cells, and importantly no difference in mast cell or basophil numbers in either untreated or TPA-treated skin (*Figure 4a* and *Figure 4—figure supplement 1a*). Similar results were found in TPA-treated *Fcer1a*$^{-/-}$ mice (*Figure 4—figure supplement 1b*). Nevertheless, qRT-PCR analysis of whole skin during TPA-induced inflammation showed that IgE-deficient mice had significantly lower levels of IL-4, IL-5, IL-6 and IL-33 but produced similar levels of TNFα and higher levels of IL-25 transcripts (*Figure 4b*). In addition, FcεRI$^+$ cells isolated from inflamed skin spontaneously produced some IL-4 and IL-13 protein, and this was further increased by IgE crosslinking (*Figure 4—figure supplement 2a,b*). A low amount of IL-5 protein was also detected (*Figure 4—figure supplement 2a,b*). Furthermore, IgE-deficient animals showed no enhancement of the enzyme histidine decarboxylase (HDC) responsible for generating histamine and significantly lower levels of the enzymes involved in the production of bioactive prostanoids such as PGD$_2$ and PGE$_2$ compared to WT mice (*Figure 4b*). Consistent with this, a significant amount of histamine was found to be released in the inflamed skin (*Figure 4c*) and into the serum (*Figure 4d*) of TPA-treated WT mice but not of TPA-treated *Igh7*$^{-/-}$ mice (*Figure 4c,d*), suggesting that degranulation and histamine discharge was dependent on IgE engagement. Skin analysis by immunohistochemistry also showed obvious changes in TPA-treated inflamed skin in the absence of IgE antibodies. Although both WT and *Igh7*$^{-/-}$ mice showed a large and equal immune infiltrate in the dermis (*Figure 4a,e*), significantly less epithelial hyperplasia was detected in the absence of IgE (*Figure 4e,f*). Together these data demonstrate that the lack of IgE substantially alters the skin microenvironment and decreases inflammation-associated epithelial hyperplasia.

## IgE activity in inflamed skin is mediated by FcεRI-signalling in basophils

To determine which cells were primarily responsible for the IgE-mediated exacerbation of skin inflammation, we sorted IgE-bearing mast cells and basophils from TPA-treated skin at a time-point when both cell populations were present. This showed that basophils produced vast amounts of cytokines such as IL-4, IL-6 and IL-13, and that they were significantly more immune active than mast cells from the same tissue (*Figure 5a* and *Figure 5—figure supplement 1*). Basophils sorted from

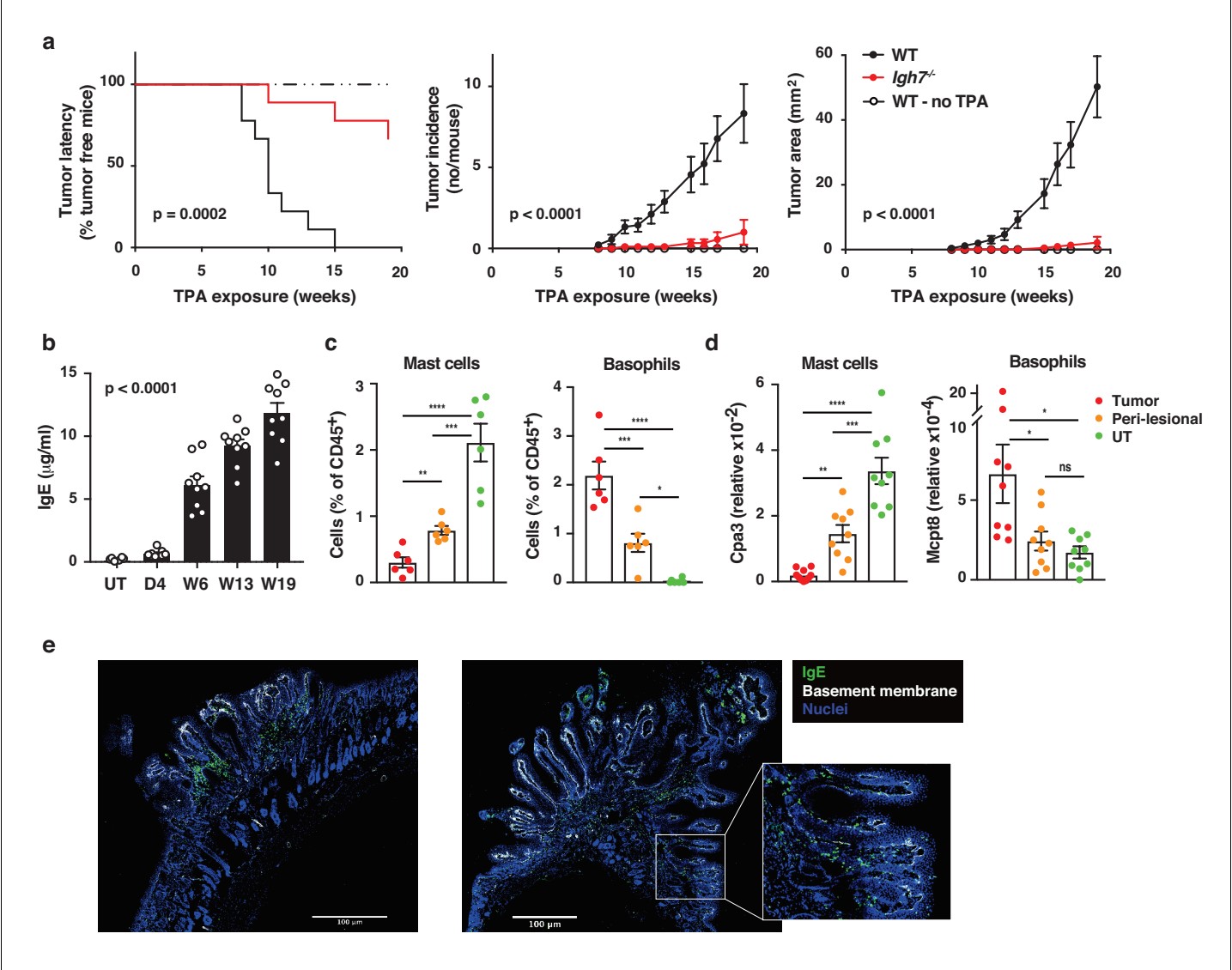

**Figure 3.** Natural IgE promotes inflammation-driven outgrowth of tumours. (**a**) Tumour susceptibility expressed as tumour latency (time to appearance of first tumour), tumour incidence (average number of tumours per mouse) and tumour area (average tumour size per mouse) in BALB/c WT and *Igh7−/−* mice (n = 9/group) mice following DMBA-TPA inflammation-driven carcinogenesis and following similar DMBA exposure without TPA (n = 6). Data are expressed as means ± SEM, and statistical significance assessed using a Log-rank (Mantel-Cox) test for tumour latency and linear regression for tumour incidence and area. (**b**) ELISA of IgE in serum of WT mice at indicated time-points during DMBA-TPA carcinogenesis (n = 9). (**c**) FACS analysis of IgE-bearing cells among the total CD45$^+$ leukocyte infiltrate in tumour tissue, peri-lesional skin and untreated (UT) belly skin of WT mice (n = 6, tumours > 3 mm were picked). Mast cells were defined as CD45$^{hi}$cKit$^+$IgE$^+$CD41$^-$ and basophils as CD45$^{lo}$cKit$^-$IgE$^+$CD41$^+$. (**d**) Quantitative RT-PCR analysis of Cpa3 transcripts (relative specific for mast cells) and Mcpt8 (relative specific for basophils) in tumour tissue, peri-lesional skin and UT belly skin from WT mice (n = 9). Data are expressed as mean ± SEM relative to the control gene cyclophilin. (**e**) Representative images of IgE staining (green) and basement membrane component CD49F (white) in inflammation (DMBA-TPA)-induced tumours. Nuclei in blue. Scale = 400 µm. Images are representative tile-scans of cross-sections from independent tumours in eight WT mice. Inset shows higher magnification of indicated area. Statistics by one-way ANOVA testing for linear trend of IgE increase with time (**b**) and one-way ANOVA multiple comparison (**c,d**); *p<0.05, **p<0.01, ***p<0.001 and ****p<0.0001. ns = not significant.

The online version of this article includes the following figure supplement(s) for figure 3:

**Figure supplement 1.** IgE-deficient mice acquire DNA-damage when exposed to DMBA but develop fewer antibodies during DMBA-TPA carcinogenesis.

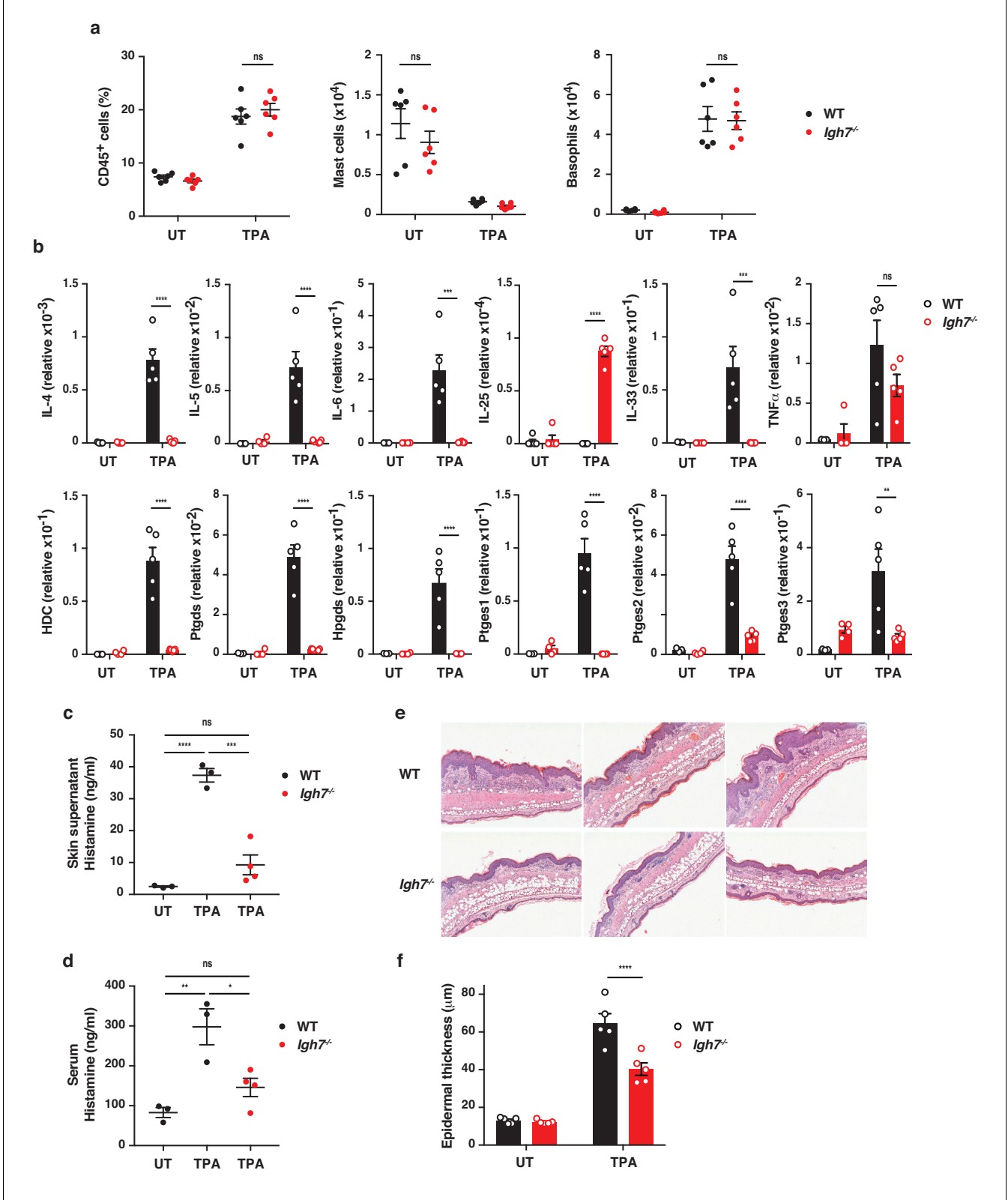

**Figure 4.** IgE alters the inflamed skin microenvironment and induces epithelial hyperplasia. (a–f) Analysis of the skin microenvironment in untreated (UT) or TPA-treated (topical TPA to the dorsal ear skin 2x/week for 2 weeks) wildtype (WT) and *Igh7⁻/⁻* mice. (a) FACS analysis of total CD45⁺ leukocytes in whole skin and enumeration of mast cells and basophil numbers (n = 6). Mast cells were defined as CD45$^{hi}$cKit⁺FcεRI⁺CD41⁻ and basophils as CD45$^{lo}$cKit⁻FcεRI⁺CD41⁺. (b) Quantitative RT-PCR analysis of the levels of the indicated transcripts in whole skin relative to that of the control gene

*Figure 4 continued on next page*

*Figure 4 continued*

cyclophilin (n = 5). (c, d) ELISA assessing the amount of histamine released ex vivo from ear skin into media when floated overnight (c) and in vivo into the serum (d) (WT n = 3, *Igh7*$^{-/-}$ n = 4). (e) Representative images of H&E-stained cross-sections from inflamed TPA-treated ear skin of WT and *Igh7*$^{-/-}$ mice. (f) Epidermal thickness is enumerated. Data in (a–d, f) are expressed as means ± SEM. One-way ANOVA multiple comparison was used to assess statistical difference; *p<0.05, **p<0.01, ***p<0.001 and ****p<0.0001. UT = untreated; ns = not significant.

The online version of this article includes the following figure supplement(s) for figure 4:

**Figure supplement 1.** The cellular immune composition of inflamed skin is unaltered in the absence of IgE-mediated signalling.

**Figure supplement 2.** FcεRI$^+$ cells from inflamed skin produce type-two cytokine proteins.

the spleen of the same animal did not express cytokine transcripts (*Figure 5b*), suggesting that the basophils only became activated when entering the inflamed skin. IgE was partly required to activate the basophils for cytokine production when entering the skin (*Figure 5a* and *Figure 5—figure supplement 1*), and IgE-signalling was necessary for degranulation and histamine release because histamine was absent from the skin and serum of both *Fcer1a*$^{-/-}$ mice (*Figure 5c*) and *Igh7*$^{-/-}$ mice (*Figure 4c,d*). Indeed, mice with a constitutive depletion of basophils resulting from Cre toxicity (*Ohnmacht et al., 2010*) (*Mcpt8*$^{Cre/+}$) showed a very limited release of histamine in inflamed skin and an absence of histamine in the serum (*Figure 5c*). These mice also demonstrated significantly less TPA-induced epithelial hyperplasia compared to WT mice (*Figure 5d*).

To investigate how IgE-mediated activation of basophils may affect skin EC, neonatal skin EC (keratinocytes) were grown in vitro and at 70% confluency supplemented with media from IgE-activated basophils or media alone as control. Basophil-derived conditioned media induced significant upregulation of Ki67 in EC, downregulation of the basal keratins K5 and K14, and upregulation of the suprabasal keratins K1 and K10 (*Figure 5e*), suggesting that basophil-released mediators promoted proliferation and upwards differentiation of skin EC. Basophil media also induced the expression of inflammatory cytokines such as IL-1α, IL-18 and IL-31 in EC (*Figure 5—figure supplement 2*). A similar pattern of induced expression of Ki67 and suprabasal keratin K1 and reduced expression of K14 was demonstrated when EC were grown in vitro with histamine (*Figure 5f*), and the dose-dependent response to histamine was completely blocked when the EC cultures were pre-incubated with histamine receptor 1 or 4 (H$_1$R or H$_4$R) antagonists (*Figure 5f*). Indeed, the basophil-induced EC proliferation and differentiation was also partially impeded by H$_1$R or H$_4$R blockade both in vitro (*Figure 5g*) and in vivo (*Figure 5h*). Thus, IgE-signalling in skin-infiltrating basophils directly promote EC activation, proliferation and differentiation, partly via the activities of histamine receptors H$_1$R and/or H$_4$R on EC.

## Basophil recruitment to the skin is dependent on TSLP/IL-3-mediated upregulation of CXCR4

Next, we explored how IgE-bearing basophils are recruited to inflamed skin. Treating mice with pertussis toxin (PTX) prior to topical TPA significantly reduced the recruitment of basophils to the skin (*Figure 6a*), indicating a role for G-protein-coupled receptors such as chemokine receptors (CCRs) in this process. In support of this idea, TPA-treated inflamed skin showed increased levels of *Ccl2*, *Ccl5*, *Ccl8*, *Ccl11* and *Cxcl12* mRNAs when compared with normal UT skin (*Figure 6b*). In parallel, infiltrating basophils showed preferential expression of *Cxcr2*, *Cxcr4* and *Ptger2* (CrTH2 – the PGD$_2$ receptor), which differed significantly from the CCRs expressed on mast cells from the same tissue (*Figure 6c*). Therefore, we tested the effect of blocking the most abundant chemokine receptors on basophil recruitment to inflamed skin. Blocking CXCR2 with a neutralising antibody inhibited the TPA-induced recruitment of neutrophils but not of basophils (*Figure 6d*), whereas mice lacking CCR2 (*Ccr2*$^{-/-}$) had reduced recruitment of monocytes but not of basophils (*Figure 6e*). Inhibition of CrTH2-signalling by COX-2 blockade or by injection of a small molecule inhibitor (AMG 853) had no effect on the recruitment of any cell type to the inflamed skin (data not shown). By contrast, blocking CXCR4 with a neutralising antibody significantly and selectively reduced basophil infiltration to the inflamed skin (*Figure 6f*), suggesting a role for CXCL12-CXCR4 in driving the basophil recruitment. However, although resting basophils abundantly expressed CXCR4, most of this was intracellular (*Figure 6g*); therefore, we explored whether any mediators that are released during skin inflammation might upregulate the surface expression of CXCR4 by basophils. Indeed, basophils

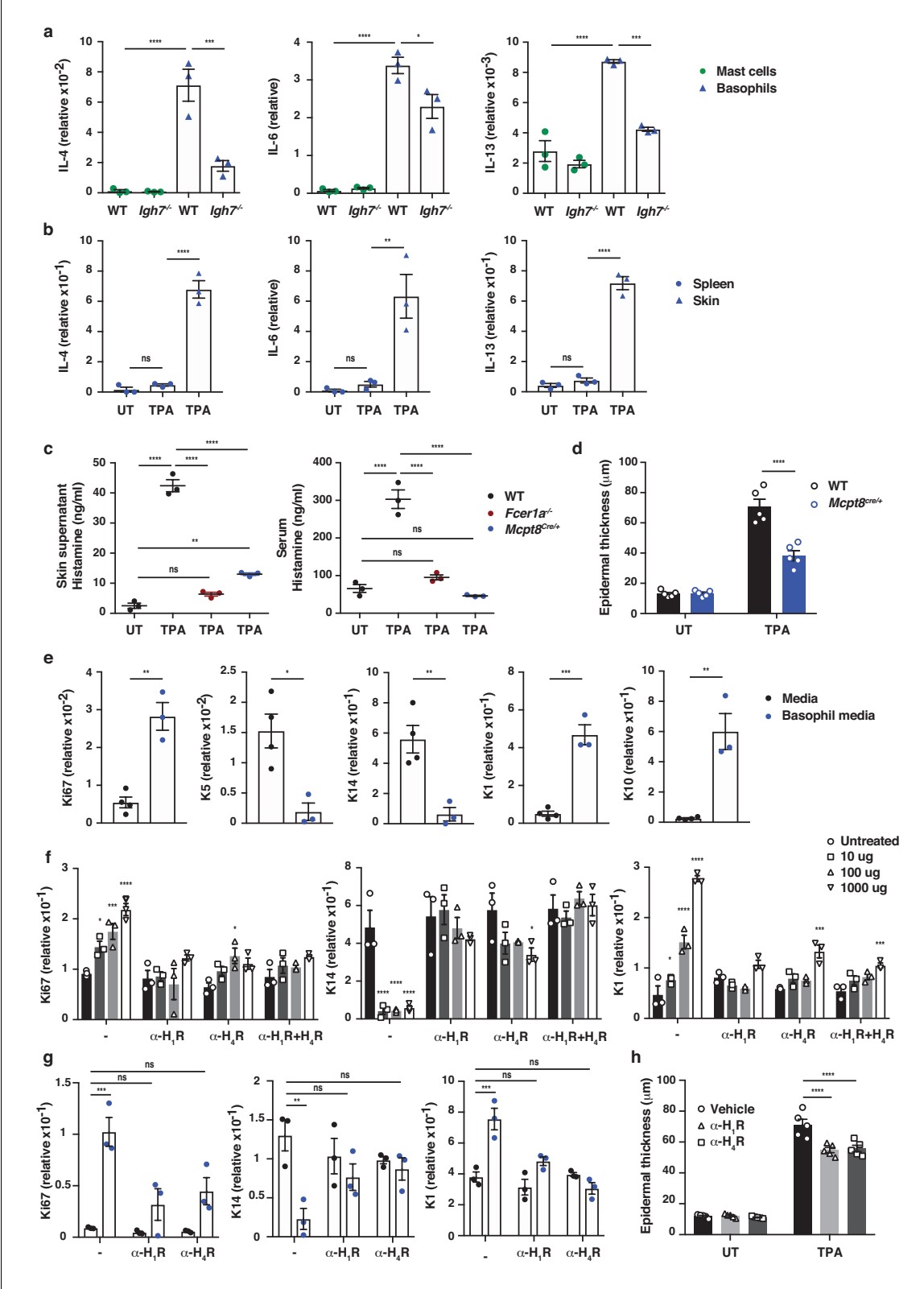

**Figure 5.** IgE activity in inflamed skin is mediated via FcεRI-signalling in basophils. (a–d, h) Healthy untreated (UT) ear skin of mice was examined and compared to inflamed TPA-treated skin (topical TPA 2x/week for 2 weeks). (a) Mast cells and basophils were FACS sorted from the skin of WT and *Igh7*⁻/⁻ mice and (b) basophils were sorted from the spleen of WT mice 72 hr after the last TPA treatment (n = 3/group). Mast cells were defined as CD45^hi^cKit⁺FcεRI⁺CD41⁻ and basophils as CD45^lo^cKit⁻FcεRI⁺CD41⁺. (a, b) Quantitative RT-PCR analysis of the levels of *Il4*, *Il6* and *Il13* transcripts in the

*Figure 5 continued on next page*

*Figure 5 continued*

FACS-sorted cell populations from the indicated tissues relative to levels of the control gene cyclophilin. (c) ELISA assessing the amount of histamine released ex vivo from ear skin into media when floated overnight (left panel) and in vivo into the serum (right panel) of WT, *FceR1a⁻/⁻* and *Mcpt8^Cre/+* mice (n = 3/group). (d) Epidermal thickness measured from H&E-stained cross-sections of WT and *Mcpt8^Cre/+* ear skin (n = 5/group). (e–g) Neonatal WT skin EC (keratinocytes) were grown in vitro to 70% confluency and then supplemented with (e) media from IgE-crosslinked basophils or media alone or (f) differing concentrations of histamine (n = 3/group). Histamine receptors on ECs were blocked by adding 10 µM $H_1R$- or $H_4R$-antagonist or both to cultures with added histamine (f) or basophil media (g) (n = 3/group). (e–g) EC were analysed by qRT-PCR for the expression levels of transcripts indicating cell cycling (Ki67) and differentiation (K5, K14, K1 and K10) relative to the expression level of the control gene cyclophilin. (h) Epidermal thickness measured from H&E-stained cross-sections of WT ear skin following in vivo blocking of $H_1R$ or $H_4R$ by i.p. administration of antagonist drug or vehicle control (n = 5/group). All data are expressed as means ± SEM. Statistical analysis was performed by one-way ANOVA multiple comparison (a–d, f–h) or two-tailed Student's t-test (e); *p<0.05, **p<0.01, ***p<0.001 and ****p<0.0001. UT = untreated; ns = not significant.

The online version of this article includes the following figure supplement(s) for figure 5:

**Figure supplement 1.** Basophils are potently immune active in inflamed skin but require IgE for full activity.
**Figure supplement 2.** IgE-activated basophils promote cytokine expression in skin EC.

that are stimulated with TSLP or IL-3, which are abundantly expressed in inflamed skin (*Figure 6h* and *Dalessandri and Strid, 2014*), (or with PMA/ionomycin as a control) showed upregulated expression of CXCR4 on the cell surface (*Figure 6i–k*). De novo expression of CXCR4 RNA was induced by TSLP (*Figure 6i*), and the intracellular pool of CXCR4 protein was transported to the cell surface when basophils were stimulated with TSLP or IL-3 (*Figure 6j,k*). As IL-3 is classically produced by T cells, we next examined the effect of depleting T cells, and found a complete lack of basophil recruitment to the inflamed skin of mice lacking all T cells (*Figure 6l*). In support, we found that reducing IL-3 levels during skin inflammation by intradermal injection of neutralising antibodies also reduced the recruitment of basophils (*Figure 6m*). Similarly, blocking TSLP significantly reduced basophil recruitment to inflamed skin and blocking TSLP and IL-3 simultaneously abolished basophil recruitment (*Figure 6m*). Injection of blocking antibodies intradermally in one ear also blocked recruitment into the inflamed skin in the contralateral ear (*Figure 6n*), suggesting that the cytokine (s) were acting systemically and not locally on basophil recruitment. In vitro experiments showed that IL-3/TSLP were not chemotactic themselves but that basophils demonstrated chemotaxis towards CXCL12 (*Figure 6—figure supplement 1*). Together these data show that production of TSLP and IL-3 in inflamed skin drives the surface expression of CXCR4 on systemic basophils, allowing recruitment to the skin in response to increased levels of CXCL12.

## FcεRI-signalling in basophils promotes inflammation-driven outgrowth of cSCCs

As natural IgE promoted EC growth via FcεRI-signalling in basophils during skin inflammation, we next tested whether a similar mechanism may support skin tumour growth. Using the two-stage inflammation-driven model of epithelial carcinogenesis, we found that mice lacking FcεRI (*Fcer1a⁻/⁻*) were significantly less susceptible to tumour development (*Figure 7a*). Only few mice developed any tumours at all and the tumours that grew were very small (*Figure 7a*). Nevertheless, the composition of the immune infiltrate in both the perilesional skin and the tumour tissue was overall similar in *Fcer1a⁻/⁻* and WT mice, apart from a minor reduction in neutrophils in the tumours from *Fcer1a⁻/⁻* mice (*Figure 7b*). Further, mice with substantially diminished FcεRI effector cells, *Cpa3^Cre/+* mice, were also significantly protected from tumour development (*Figure 7c*). *Cpa3^Cre/+* mice lack all mature skin mast cells as a result of Cre toxicity (*Feyerabend et al., 2011*) and no mast cells were found in either perilesional skin or tumour tissue (*Figure 7d*). However, these mice also have substantially reduced basophil numbers with >90% reduction in inflamed skin and perilesional skin, although basophils were detected in the tumour (*Figure 7d*). Owing to the dual effect of the *Cpa3^Cre* on mast cells and basophils, we next used *Mcpt8^Cre/+* mice, which have normal mast cell numbers but a strongly reduced number of basophils (*Figure 7f*). The *Mcpt8^Cre/+* mice were again significantly less susceptible to tumour growth (*Figure 7e*), suggesting that FcεRI-signalling in basophils was responsible for tumour growth. As signalling via histamine receptors $H_1R$ and/or $H_4R$ on EC were shown to promote EC proliferation and differentiation during skin inflammation, we then tested whether blocking $H_1R$ affected tumour outgrowth. Indeed, adding cetirizine, a $H_1R$ blocker, to the drinking water during tumour promotion significantly delayed tumour onset and

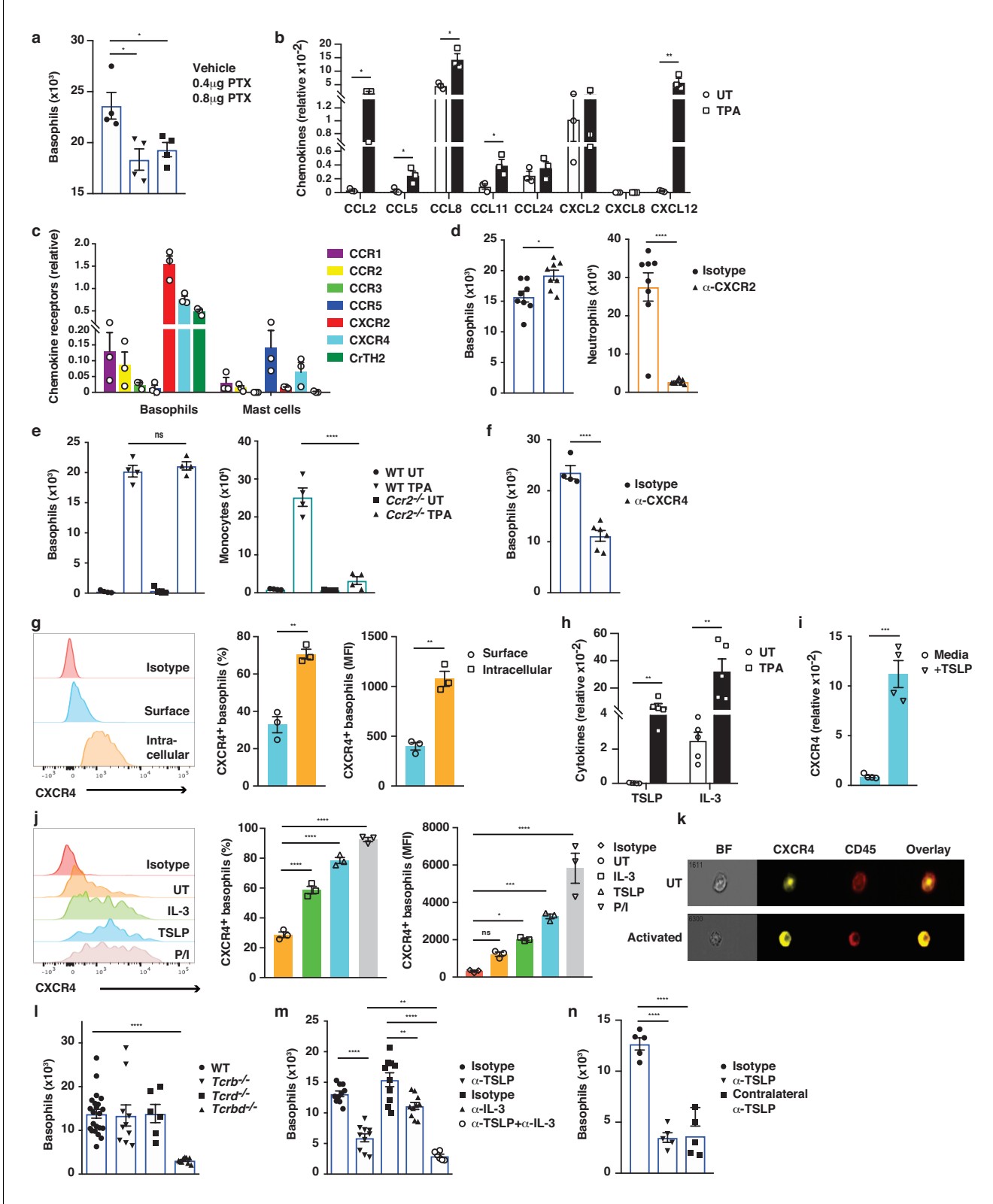

**Figure 6.** Basophil recruitment to the skin requires TSLP/IL-3-mediated upregulation of CXCR4. (**a–f, h, l–n**) Inflamed skin was analysed 24 hr after a single topical application of 2.5 nM TPA and compared to healthy UT skin. (**a**) Recruitment of basophils to inflamed skin was analysed by FACS and presented as absolute numbers of basophils in ear skin following i.v. injection of PTX or vehicle control. Basophils were gated as CD45$^{lo}$cKit$^-$IgE$^+$CD41$^+$. (**b**) Expression of chemokines in whole skin and (**c**) expression of chemokine receptors in sorted skin basophils and mast cells

*Figure 6 continued on next page*

*Figure 6 continued*

were analysed by qRT-PCR and presented relative to the expression of the control gene cyclophilin. (d–f) Recruitment of basophils to inflamed skin was analysed as in panel (a): (d) after i.v. injection of 50 ng anti-CXCR2 antibody, (e) in *Ccr2*$^{-/-}$ mice, or (f) after i.v. injection of 50 ng anti-CXCR4 antibody. Suitable control populations were used to show the efficacy of a treatment where appropriate. (g, i–k) Basophils were isolated from the spleen of WT mice and (g) analysed for cell surface or intracellular expression of CXCR4 by flow cytometry. (h) Expression of IL-3 and TSLP in whole inflamed and untreated (UT) skin was analysed by qRT-PCR and presented relative to the expression of the control gene cyclophilin. (i) Splenic basophils were incubated with 1 µg/ml TSLP or media alone for 24 hr, and expression levels of CXCR4 transcripts were assessed by qRT-PCR. (j, k) Basophils were activated for 24 hr with 100 ng/ml IL-3, 1 µg/ml TSLP or isotype control, with PMA/ionomycin for 4 hr, or left UT before surface expression of CXCR4 was analysed by (j) FACS or (k) total CXCR4 expression (yellow) imaged with surface staining of CD45 (red) using Imagestream. (l–n) Recruitment of basophils to inflamed skin was analysed as in panel (a): (l) in *Tcrb*$^{-/-}$, *Tcrd*$^{-/-}$ and *Tcrbd*$^{-/-}$ mice, (m) following intradermal (i.d.) injection of 20 ng anti-TSLP or anti-IL-3 (or both) antibodies into the TPA treated ear or (n) into the contralateral ear. Representative data are shown in panels (g, j [left panels]) as histograms of CXCR4 FACS stain and in panel (k) as images from Imagestream; all other data are expressed as means ± SEM. Statistical analyses were performed by one-way ANOVA multiple comparison (a–c, e, g, j, l–n) or two-tailed Student's t-test (d, f, h–i); *p<0.05, **p<0.01, ***p<0.001 and ****p<0.0001. UT = untreated; ns = not significant; BF = bright field.

The online version of this article includes the following figure supplement(s) for figure 6:

**Figure supplement 1.** Basophils demonstrate chemotaxis towards CXCL12.

suppressed tumour growth (*Figure 7h*). Together these data demonstrate that the inflammation-driven tumour promotion is mediated via FcεRI-signalling in skin-infiltrating basophils, partly via histamine release.

## Discussion

In this study, we found that skin inflammation enhanced levels of natural IgE, which accumulated in the tissue mainly on infiltrating basophils recruited via TSLP/IL-3-dependent expression of CXCR4. IgE-activated basophils promoted EC growth and differentiation, and strongly drove the outgrowth of inflammation-driven skin SCCs, partly via histamine engagement of H$_1$R and H$_4$R on EC.

Natural IgE is present at low concentrations in the blood of mice and men but is constitutively available on FcεRI-expressing cells at our barrier surfaces, such as the skin. Although the role of natural IgE at steady state has not been widely studied, it is suggested to be part of our 'frontline' immune defence against a variety of environmental challenges (such as parasitic infections) as well as exposure to noxious xenobiotics, irritants and venoms (*Palm et al., 2012*; *Profet, 1991*). The host defences that are evoked may include barrier enhancement, removal or inactivation of the challenge and tissue repair (*Palm et al., 2012*; *Profet, 1991*). Our IgE sequencing analysis revealed that IgE antibodies that are induced by low-grade tissue inflammation have a similar repertoire and VDJ gene-usage to the natural IgE in healthy animals, suggesting that tissue inflammation merely enhances the level of IgE already present in the host. This natural IgE fortified the skin barrier defences by inducing EC growth and differentiation and by thickening the epidermis. However, this response should only be activated transiently upon exposure to noxious stimuli, because long-term exacerbation of this response may lead to pathologies and altered EC function. Indeed, we found that when EC harbour oncogenic mutations, chronic activation of the IgE response subverts its protective effects and supports the growth of otherwise subclinical EC lesions, thereby promoting tumour development.

The tumour-promoting effect of IgE in the context of chronic skin inflammation is in stark contrast to our recently published data showing that IgE provides protection against mutagen-driven EC carcinogenesis (*Crawford et al., 2018*). Interestingly, the DNA-damage caused by repeated carcinogen exposure drives a de novo induction of autoreactive IgE antibodies, which differ substantially in their repertoire from the inflammation-induced IgE (*Crawford et al., 2018*). Repeated carcinogen exposure also gives rise to substantially more mutated tumours than inflammation-driven skin carcinogenesis (*Nassar et al., 2015*), suggesting a different mode of tumour outgrowth in the different experimental scenarios. Whole-exome sequencing of human cutaneous squamous-cell carcinoma (cSCC) has identified this as one of the most mutated human cancers, with a mean of 50 mutations per megabase of DNA (*Inman et al., 2018*). This sequencing has also revealed that there are significantly more mutations in immune-related pathways, including in FcεRI-signalling, in tumours of high-versus low-risk of metastasis (*Inman et al., 2018*). We also found a significant positive correlation

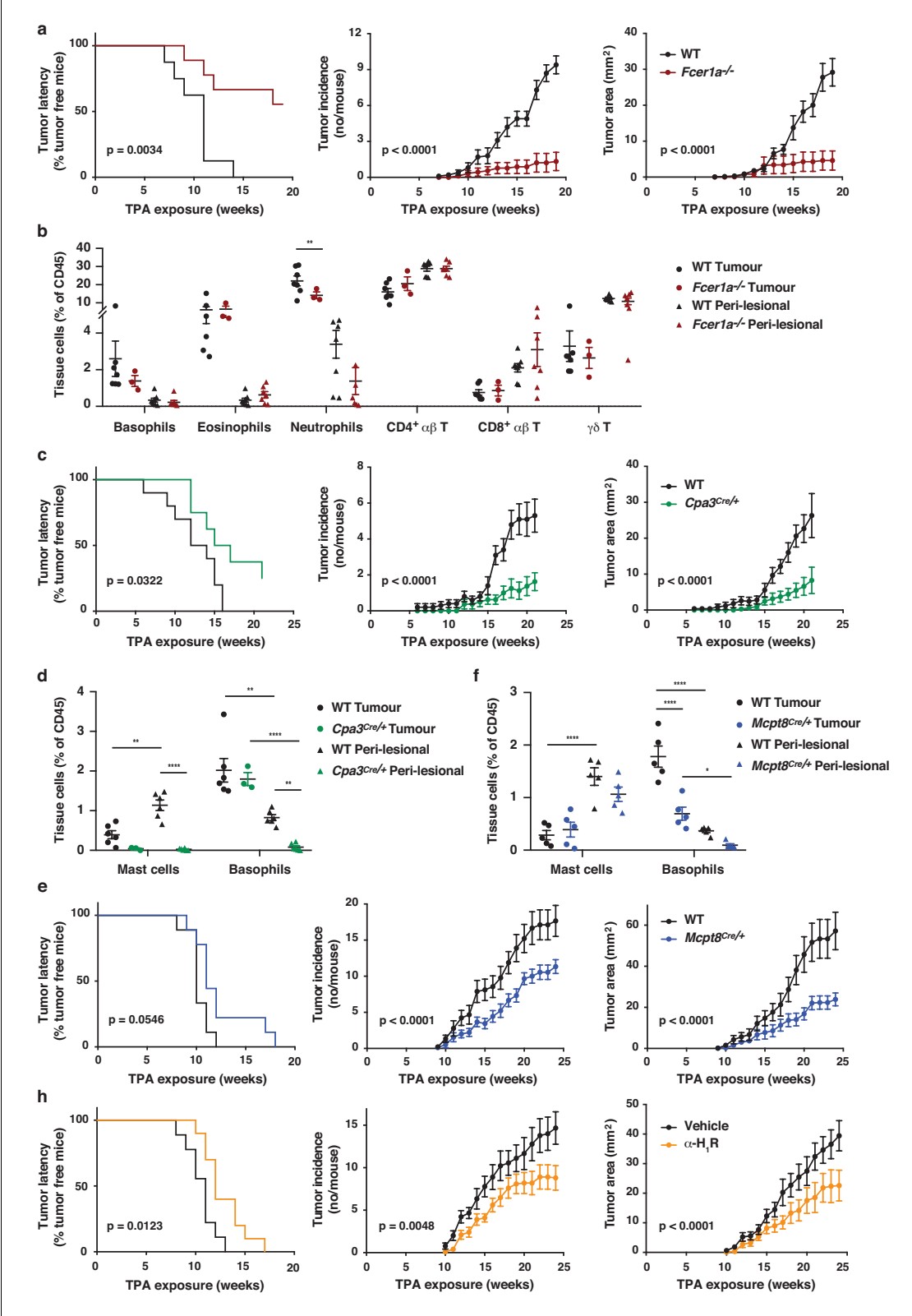

**Figure 7.** FcεRI-signalling in basophils promotes inflammation-driven outgrowth of cSCCs. (a–h) Mice were treated with a single subclinical dose of DMBA to the shaved dorsal back skin, and skin inflammation was promoted by twice weekly TPA application for 18–25 weeks. Tumour susceptibility is expressed as tumour latency (time to appearance of first tumour), tumour incidence (average number of tumours per mouse) and tumour area (average tumour size per mouse) in (a) WT (n = 10) and *FceR1a*$^{-/-}$ mice (n = 9), (c) WT (n = 10) and *Cpa3*$^{Cre/+}$ mice (n = 8), (e) WT and *Mcpt8*$^{Cre/+}$ mice (n = 9/

*Figure 7 continued on next page*

*Figure 7 continued*

group) and (h) WT mice given the $H_1R$ antagonist cetirizine at 30 mg/kg or vehicle control in the drinking water throughout the experiment (n = 10/ group). (b) FACS analysis of leukocyte infiltrate in tumour tissue and perilesional skin of WT and $FceR1a^{-/-}$ mice (n = 7 for WT groups; n = 3 for tumour, n = 7 for peri-lesional tissue for $FceR1a^{-/-}$) presented as proportion of total $CD45^+$ leukocytes. (d, f) FACS analysis of mast cells and basophils in tumours and peri-lesional tissue of (d) WT and $Cpa3^{Cre/+}$ mice (n = 6 for WT groups; n = 3 for tumour, n = 6 for peri-lesional tissue for $Cpa3^{Cre/+}$) and of (f) WT and $Mcpt8^{Cre/+}$ mice (n = 5/group). Data are expressed as means ± SEM, and statistical significance was assessed using Log-rank (Mantel-Cox) test for tumour latency and linear regression for tumour incidence and area (a, c, e, h) and one-way ANOVA multiple comparison (b, d, f); p<0.05, **p<0.01 and ****p<0.0001.

between *FCER1A*-expression in human cSCC and good disease prognosis (*Crawford et al., 2018*), which may reflect a broader paradigm as *FCER1A*-expression correlates with overall prolonged survival in many human malignancies (*Gentles et al., 2015*). Thus, although the commonly used inflammation-driven cutaneous carcinogenesis mouse model (DMBA-TPA; also used in this study) exhibits recurrent mutations in genes similar to those mutated in human cSCC (*Hras, Kras, Rras2* or *Trp53*) (*Nassar et al., 2015*), the low level of mutations may alter the immunogenicity and the microenvironmental niche in which the tumour grows, which may in part explain the paradoxical role of IgE in this model. Together, our previous (*Crawford et al., 2018*) and current studies show a potent link between IgE and cancer. However, the biological consequences of IgE engagement clearly depend on the nature of the antibodies and the tissue microenvironment.

Chronic inflammation has frequently been linked to cancer promotion, and tumour extrinsic inflammatory conditions may be key to the development of selected cancers. Epidemiological studies suggest that low-grade tissue inflammation predisposes tissues to different forms of cancer by contributing to the proliferation and survival of malignant cells, by promoting angiogenesis and tumour metastasis and by subverting anti-cancer immunity (*Mantovani et al., 2008*; *Coussens et al., 2013*; *Grivennikov et al., 2010*). For example, gastric and bladder cancers have long been known to be driven by inflammation (*Fox and Wang, 2007*; *Echizen et al., 2019*; *Michaud, 2007*). Interestingly, RNAseq data in these two cancers, contrary to that in many other cancers, suggests a negative correlation between *FCER1A* and overall survival (KMplotter; *Nagy et al., 2018*). A recent study also revealed increased mast cell density in human gastric tumours and in the corresponding mouse models, and showed that mast cell degranulation was driving tumour growth (*Eissmann et al., 2019*). It should be noted that in this study mast cell numeration was performed using toluidine blue, which also stains basophils. Although IgE/FcεRI-signalling has not been linked directly to tumour growth in these cancers, taken together, these data suggest that IgE may play a role in cancer promotion in inflammatory settings in some human tissues.

IgE-mediated immune surveillance in the skin is most probably dominated by mast cells in healthy skin and during the early-phase response (as is immune surveillance in allergic disease), whereas the late-phase response is driven by recruited basophils. In our study, basophils were dominant over mast cells in the inflamed skin, and only basophils appeared to enter the tumours, whilst mast cells remained at the base of the tumours and in the surrounding tissue. Epidermal hyperplasia during tissue inflammation was driven by FcεRI-signalling in basophils, and tumour growth also appeared to be promoted mainly by basophils, although we cannot exclude a role for mast cells because *Cpa3-Cre/+* mice (which lack mast cells and most basophils) were more protected from carcinogenesis than *Mcpt8^{Cre/+}* mice (which have normal mast cell numbers but lack most basophils). Further, mast cells have previously been shown to support SCC growth in tumour models driven by human papilloma virus (*Coussens et al., 1999*; *Bergot et al., 2014*) and indeed may contribute to the tumour promotion shown in our study. Nevertheless, other skin cancer studies have not found a clear role for mast cells in tumour progression or even shown an anti-tumourogenic role; thus, the role of mast cells in skin cancer development remain controversial and most probably depends on the underlying factors driving the disease (*Varricchi et al., 2017*). Although the physiological role of basophils remain poorly understood, it is increasingly recognised that basophils regulate immunity at many levels, often distinct from mast cells (*Voehringer, 2013*; *Karasuyama et al., 2009*; *Schwartz et al., 2016*). For example, skin-infiltrating basophils, but not mast cells, have been shown to be crucial for acquired tick resistance in mice (*Tabakawa et al., 2018*). Interestingly, the mechanism of basophil-mediated tick host defence is by epidermal thickening driven by local histamine release

(*Tabakawa et al., 2018*), consistent with the role of basophils and histamine in EC growth demonstrated in our study.

Histamine has wide-ranging effects on skin ECs and has previously been shown to promote wound healing in the skin (*Gutowska-Owsiak et al., 2014*). For a long time, the effects of histamine in skin diseases have been thought to be mediated solely by its action on $H_1Rs$, however newer findings, including this report, suggest that $H_4R$-signalling strongly affects skin ECs. $H_4R$-deficient mice have thinner epidermis than WT mice at steady state (*Glatzer et al., 2013*). Furthermore, $H_4R$-deficiency or blockade significantly reduces epidermal hyperplasia (*Rossbach et al., 2016*) and total skin thickness (*Cowden et al., 2010*) in inflammatory settings. Surprisingly, mice lacking histidine decarboxylase (HDC) (the enzyme responsible for histamine generation) are more susceptible to inflammation-induced carcinogenesis of the skin and colon (*Yang et al., 2011*). However, this is due to an important role for histamine in myeloid cell differentiation and maturation within the bone marrow (through $H_1R$ and $H_2R$), and HDC-deficient mice have significantly altered myeloid cell composition including a large increase in immature myeloid cells, which are recruited to the tissue during inflammation and promote cancer growth (*Yang et al., 2011*).

In summary, we demonstrate how IgE antibodies with natural specificities regulate skin EC growth and differentiation via FcεRI-signalling. Further, during chronic skin inflammation, IgE strongly promotes outgrowth of ECs harbouring oncogenic mutations. Thus, our data suggest a potential link between autoreactive IgE and skin hyperplasia, which may be pertinent for a variety of inflammatory skin conditions in which the blocking of IgE-mediated signalling could be beneficial, as has been shown for idiopathic urticaria (*Maurer et al., 2011*; *Maurer et al., 2013*). Moreover, our data may have potential important implications for both atopy and cancer.

## Materials and methods

### Mice

Genetically altered mice were generated as previously described. $Igh-7^{-/-}$ (*Oettgen et al., 1994*), $Fcer1a^{-/-}$ (*Dombrowicz et al., 1993*), $Cpa3^{Cre/+}$ (*Feyerabend et al., 2011*) and $Mcpt8^{Cre/+}$ (*Ohnmacht et al., 2010*) were on the BALB/c background after more than 10 backcrosses. $Tcrd^{-/-}$ (*Itohara et al., 1993*) and $Tcrb^{-/-}$ (*Mombaerts et al., 1991*) were on the FVB/N background after more than 10 backcrosses and $Tcrdb^{-/-}$ were generated by intercrossing the respective mutants. $Ccr2^{-/-}$ (*Boring et al., 1997*) mice were on the CD1 background. Non-transgenic littermates were used as controls or strain-matched wild-type control animals were purchased from Charles River. Mice were bred and maintained in individually ventilated cages under specific pathogen-free conditions. Age-matched, female mice were used for all experiments at $\geq 7$ weeks of age and selected at random from a large pool when allocated to experiments. All studies were approved by Imperial College AWERB (Animal Welfare and Ethical Review Body) and by the UK Home Office. Experiments involving cancer studies strictly adhered to the guidelines set out by the National Cancer Research Institute (NCRI) and by *Workman et al. (2010)*, 'Guidelines for the Welfare and Use of Animals in Cancer Research'. All studies using animals were conducted following the Animal Research: Reporting In Vivo Experiments (ARRIVE) guidelines (*Kilkenny et al., 2010*).

### Cutaneous challenge and chemical carcinogenesis

Chemicals 12–0-tetradecanoylphorbol-13-acetate (TPA), MC903 and 7,12-dimethylbenz[a]anthracene (DMBA) were purchased from Sigma, R848 from Enzo Life Sciences and they were dissolved in ethanol (TPA, MC903) or acetone (DMBA, R848). Skin inflammation was induced by exposing the dorsal sides of the ear skin to a single or repeated dose of 2.5 nM TPA, 1 nM MC903 or 100 μg R848 in 25 μl.

For cutaneous carcinogenesis, age-matched female mice were used at 7 weeks of age. The fur of the dorsal back area was removed using hair clippers and mice were rested for 1 week. Applications of chemicals and tumour monitoring were performed as previously described (*Strid et al., 2008*). In brief, 600 nM DMBA was carefully and slowly applied by pipette, in a 150 μl volume, to the entire clipped skin area. Mice were rested for one week and 20 nM TPA was then applied twice weekly. Hair regrowth during the experiment was gently removed by clipping with trimmers. Mice were monitored daily and cutaneous tumours were counted and measured with a calliper once weekly.

Back skin and tumours were evaluated by visual inspection by an observer blinded to the experimental groups.

## Tissue processing

Skin and tumour tissues were cut into small 1 mm$^2$ pieces using a scalpel blade and incubated for 2 hr in digestion buffer containing 25 ug/ml liberase (Roche), 250 ug/ml DNAseI (Roche) and 1x DNAse buffer (1.21 Tris base, 0.5 g MgCl$_2$ and 0.073 g CaCl$_2$) at 37˚. Following digestion, tissue was transferred into C-tubes (Miltenyi Biotech) containing RPMI-1640 media (Thermo Fisher) supplemented with 10% heat-inactivated foetal calf serum, 1% penicillin-streptomycin-glutamine (Thermo Fisher) and physically disrupted using a Miltenyi cell dissociator. Spleen and LN tissue was disaggregated mechanically using a rubber policeman. All cell suspensions were filtered and cells were counted using a CASY cell counter (Roche).

## Dermal sheets

Ears were collected, split into dorsal and ventral sides and floated dermis side down in 0.5 M NH4SCN for 40 min, 37˚C and 5% v/v CO$_2$. Subsequently, intact dermal sheets were isolated and washed in PBS. They were then fixed in ice-cold acetone at –20˚C for 15 min, followed by rehydration in PBS and blocking of nonspecific binding with 2% BSA for 1 hr. IgE in the tissue was visualised by first incubating with rat anti-mouse IgE (R35-92, BD) followed by Alexa Fluor 555–conjugated goat anti-rat IgE (A21434, ThermoFisher), each for 1 hr at room temperature. After extensive washing in PBS, samples were mounted in VectaShield containing DAPI (Vectashield) and visualised with a Leica SP5 confocal laser-scanning microscope (Leica).

## Epidermal thickness

The dorsal sides of the ears were treated topically twice a week for 3 weeks with 2.5 nM TPA and subsequently fixed and paraffin embedded. Ear skin from untreated mice or the untreated contralateral ear was embedded in parallel. 5 μm sections were cut and stained for H&E. Images were obtained using a Leica DM4B microscope (Leica). Epidermal thickness was measured using Fiji software by taking five measurements per section from five sections per ear in a blinded manner. In some experiments, H$_1$R or H$_4$R antagonist was administered by i.p. injection of 20 mg/kg cetirizine dihydrochloride (Tocris) or 20 mg/kg JNJ 7777120 (Tocris), respectively, or of the appropriate vehicle controls.

## Immunofluorescent staining of skin and tumour samples

For ear skin, ears were removed and a defined central section cut. Tumours were removed from the back of the ears along with a small piece of adjacent skin. All tissues were snap-frozen in OCT on dry ice. 6 μm sections were cut using a Leica JUNG CM1800 cryostat and stored at –80˚C.

For staining, slides were returned to room temperature before fixation with 4% paraformaldhyde for 15 min. Samples were then blocked in 5% goat serum for 1 hr at room temperature before staining with rat anti-mouse IgE (R35-92, PharMingen) overnight at 4˚C. Following staining, samples were washed and incubated with Alexa Fluor 555–conjugated goat anti-rat IgG (A21434, ThermoFisher). After extensive washing, samples were then either further incubated with Alexa Flour 647-conjugated rat anti-mouse CD49F (GoH3, BioLegend) or directly mounted in VectaShield containing DAPI (Vectashield). Tissue samples were visualised with a Leica SP5 confocal laser-scanning microscope (Leica).

## Flow cytometry, cell sorting and Imagestream

Cell suspensions were blocked for non-specific binding using antibody against FcγR (2.4G2) and 2% normal rat serum (Sigma) prior to any staining protocols. For staining of cell surface markers, cell suspensions were stained with fluorochrome-conjugated antibodies or appropriate isotype control with the addition of a fixable, live/dead discrimination dye (Invitrogen) for 25 min and subsequently washed. Intracellular staining was then carried out using an Intracellular Staining kit as per manufacturer's instructions (ThermoFisher Scientific). Cells were fixed for 10 min at 4˚C, washed with Perm buffer and stained for intracellular markers for 25 min. For intranuclear γH2AX staining, cells were fixed/permeablised in ice-cold 70% ethanol at −20˚C for 2 hr, blocked with 2% normal mouse serum

(Sigma), Fc-block and 2% fetal calf serum for 15 min, followed by 45 min staining for γH2AX at room temperature. Stained cells were analysed using BD FACSVerse and Fortessa X20 (BD Biosciences, NJ, USA) machines. Data analysis was performed using FlowJo 10 for Mac (TreeStar, OR, USA). For cell sorting, a FACS AriaIII High Speed Cell Sorter (BD) was used. For Imagestream analysis, internalisation scores were determined using CD45 staining as a membrane marker and CXCR4 staining as the probe. Analyses were performed using an ImageStream X Mark II imaging flow cytometer and IDEAS v6 software (AMNIS).

Antibodies were sourced from Biolegend unless otherwise stated. The following antibodies were used: anti-CD11b (M1/70), anti-Siglec-F (E50-2440), anti-Ly6C (HK1.4), anti-Ly6G (IA8), anti-CD4 (GK1.5), anti-CD8 (53–6.7), anti-CXCR4 (I.276F12), anti-CD138 (281–2, BD) anti-CD45 (30-F11, eBioscience), anti-IgE (23G3, eBioscience), anti-TCRβ (H57-597), anti-TCRγδ (eBioGL3, eBioscience), anti-Vγ5 (536, eBioscience), anti-Vγ4 (UC3-10Ab), anti-CD117 (2B8), anti-CD41 (MWReg30), anti-FcεRI (MAR-1), anti-CD95 (Jo2, eBioscience), γH2AX (JBW301, Millipore), anti-IL-4 (11B11, eBioscience), anti-IL-5 (TRFK5) and anti-IL-13 (eBio13A, eBioscience).

Cells subsets were gated on live singlets for all and then as: mast cells — $CD45^+CD117^+IgE/FcεRI^+CD41^-$; basophils — $CD45^{mid}CD117^-IgE/FcεRI^+CD41^+$; eosinophils — $CD45^+GR1^+CD11b^+$-$Siglec-F^+$; neutrophils — $CD45^+Ly6G^+Ly6C^-CD11b^+$; monocytes/macrophages — $CD45^+CD11b^+$-$Ly6G^-Ly6C^+$; αβT cells — $CD45^+Tcrb^+CD4/CD8^+$; γδ IEL — $CD45^+Tcrd^{hi}Vg5^+$; dermal γδ T cell — $CD45^+Tcrd^+Vg5^-$; plasma cell — $FSC^{hi}CD95^+CD138^+$.

## ELISA

Total IgE was measured by an IgE capture method. Sera to be tested and IgE standard were added to plate wells coated with 1 µg/ml rat monoclonal anti-mouse IgE (PharMingen) and blocked with 1% rat serum. Biotinylated rat monoclonal anti-mouse IgE (PharMingen) at 1 µg/ml were then added and incubated for 2 hr at 37°C. After washing in PBS 0.05% Tween, plates were incubated with alkaline phosphatase-conjugated Streptavidin (PharMingen) for 1 hr at 37°C. After washing, detection of antibody levels was carried out by addition of alkaline phosphatase substrate pNPP (Sigma) and absorbance was measured at 405 nm. For detection of anti-phosphorylcholine (anti-PC) specific IgE, total IgE was captured with anti-mouse IgE (PharMingen), followed by addition of PC-BSA at 50 mg/ml (Sigma). After washing, detection was achieved by incubating first with anti-PC IgM antibody (clone BH8) (Millipore), followed by anti-IgM conjugated to alkaline phosphatase (Southern Biotech) and addition of pNPP substrate (Sigma). Absorbance was measured at 405 nm. Detection limit was 3 ng/ml.

For IgG1 and IgG2a antibodies, NUNC Immune Maxisorp 96-well plates (Thermo Scientific) were coated with 5 µg/ml goat anti-mouse IgH+L (Southern Biotech) in borate buffered saline at 37°C for 3 hr. After washing, plates were blocked with PBS containing 0.5% BSA for 1 hr at room temperature, and appropriately diluted sera added and incubated overnight at 4°C. After washing, plates were incubated with alkaline phosphatase-conjugated polyclonal goat anti-mouse IgG1 or IgG2a (which also detects IgG2c in C57BL/6 mice) (both Southern Biotech) for 5 hr at 4°C. Following further washing, the alkaline phosphatase substrate pNPP (Sigma) was added and absorbance measured at 405 nm.

Histamine levels in serum and ear supernatant were measured using an ultra-sensitive histamine ELISA kit (Enzo Life Sciences) following the manufacturer's instructions. Skin supernatant was acquired by floating the dorsal ear skin on 250 µl of complete media for 16 hr.

## qRT-PCR and primer sequences

RNA was extracted from isolated cell suspensions or from skin/tumour tissue, and preserved in RNA-later, using RNEasy Mini kits (Qiagen). RNA was dissolved in nuclease-free water, and yield and purity were determined. Complementary DNA (cDNA) was synthesised from RNA with an iScript cDNA synthesis kit (Bio-Rad) as per the manufacturer's instructions. cDNA was diluted in nuclease-free double-deionized water for qRT–PCR. All primers were single-stranded DNA oligonucleotides (Sigma) that were intron-spanning, as verified by NCBI Primer-Blast tool. Real-time PCR products were detected with SYBR Green (Life) measured continuously with a ViiA 7 Real-Time PCR system (Applied Biosystems, CA, USA). Ct values for genes of interest were normalised against Ct values of the housekeeping gene Cyclophilin (Cyc) using the $2^{-\Delta Ct}$ method. The following primers were

used — IL-1α: F (5′- TTGGTTAAATGACCTGCAACA-3′), R (5′-GAGCGCTCACGAACAGTTG-3′), IL-4: F (5′-CATCGGCATTTTGAACGAG-3′), R (5′-CGAGCTCACTCTCTGTGGTG-3′), IL-5: F (5′-GAAAGA-GACCTTGACACAGCTG-3′), R (5′-GAACTCTTGCAGGTAATCCAGG-3′), IL-6: F (5′-TGATGGATGC TACCAAACTGG-3′), R (5′-TTCATGTACTCCAGGTAGCTATGG-3′), IL-13: (5′-ACACAAGACCAGAC TCCCC-3′), R (5′-CTCCTCATTAGAAGGGGCCG-3′), IL-18: F (5′-ACATCTTCTGCAACCTCCAGCA-3′), R (5′-CATTGTTCCTGGGCCAAGAGG-3′), IL-25: (5′- TGGAGCTCTGCATCTGTGTC-3′), R (5′-GA TTCAAGTCCCTGTCCAACTC-3′), IL-31: F (5′-GGCCTTCCTCACTCTCTTAC-3′), R (5′-GTATAG-GAACCTGGCTGGC-3′), IL-33: F (5′-CACATTGAGCATCCAAGGAA-3′), R (5′-AACAGATTGGTCA TTGTATGTACTCAG-3′), TNFα: F (5′-AGCCCACGTAGCAAACCACCA-3′), R (5′-ACACCCATTCCC TTCACAGAGCAAT-3′), Ki67: F (5′-TCTGATGTTAGGTGTTTGAG-3′), R (5′-CACTTTTCTGGTAACTTC TTG-3′), K1: F (5′-TTTGCCTCCTTCATCGACA-3′), R (5′-GTTTTGGGTCCGGGTTGT-3′), K5: F (5′-CC TGCAGAAGGCCAAGCA-3′), R (5′-TGGTGTTCATGAGCTCCTGGTA-3′), K10: F (5′-GGATGAGC TGACCCTTAGCA-3′), R (5′-CATTTTGAAGGTCTCTCATTTCCT-3′), K14: F (5′-CAGCCCCTAC TTCAAGACCA-3′), R (5′-GGCTCTCAATCTGCATCTCC-3′), Cpa3: F (5′-CTACGGCCCAATAGCA TCCA-3′), R (5′-TGCCCAGGTCATAAACCCAG-3′), Mcpt8: F (5′-CAGTCTATCGCTGTGGTGGT-3′), R (5′-GAGCTTTGCGTTCCAGCTTC-3′), HDC: F (5′-GAGTGCACAGCACAGACAAAGG-3′), R (5′-TC TAGCTCGGTAGTATTCACT-3′), Ptgds: F (5′-GACACAGTGCAGCCCAACTTTC-3′), R (5′-GGGCTAC-CACTGTCTTGCACATA-3′), Hpgds: F (5′-ATCCACCAGAGCCTCGCAATAG-3′), R (5′-TCATCCAGCG TGTCCACCA-3′), Ptges1: F (5′-GGATGCGCTGAAACGTGGA-3′), R (5′-CAGGAATGAGTACAC-GAAGCC-3′), Ptges2: F (5′-CTCATCAGCAAGCGCCTCAA-3′), R (5′-GGTCTTTACCCACGGCTGTCA-3′), Ptges3: F (5′-ATCACATGGGTGGTGATGAGGA-3′), R (5′-AGGCGATGACAACAGCCCTTAC-3′), CCL2: F (5′-CCCAATGAGTAGGCTGGAGA-3′), R (5′-AAAATGGATCCACACCTTGC-3′), CCL5: F (5′-TGCCCACGTCAAGGAGTATTTC-3′), R (5′-AACCCACTTCTTCTCTGGGTTG-3′), CCL8: F (5′-CCC TTCGGGTGCTGAAAAG-3′), R (5′-TCTGGAAAACCACAGCTTCCA-3′), CCL11: F (5′-ATGCACCC TGAAAGCCATAGTC-3′), R (5′-CAGGTGCTTTGTGGCATCCT-3′), CCL24: F (5′-CGGCCTCCTTCTCC TGGTA-3′), R (5′-TGGCCAACTGGTAGCTAACCA-3′), CXCL2: F (5′-CGCTGTCAATGCCTGAAG-3′), R (5′-GGCGTCACACTCAAGCTCT-3′), CXCL8: F (5′-CTCTTGGCAGCCTTCCTGATT-3′), R (5′-TA TGCACTGACATCTAAGTTCTTTAGCA-3′), CXCL12: F (5′-GAGCCAACGTCAAGCATCTG-3′), R (5′-CGGGTCAATGCACACTTGTC-3′), CCR1: F (5′-AAGGCCCAGAAACAAAGTCT-3′), R (5′-TCTGTAG TTGTGGGGTAGGC-3′), CCR2: F (5′-ACACCCTGTTTCGCTGTAGG-3′), R (5′-GATTCCTGGAAGG TGGTCAA-3′), CCR3: F (5′-ATGGCATGTGTAAGCTCCTCTCAG-3′), R (5′-TTGCTCCGCTCACAGTCA TTTCCCCR-3′), CXCR2: F (5′-AGCAAACACCTCTACTACCCTCTA-3′), R (5′-GGGCTGCATCAA TTCAAATACCA-3′), CXCR4: F (5′-TCAACCTCTACAGCAGCGTTCTCTT-3′), R (5′-TGTTGGTGGCG TGGACAAT-3′), CrTH2: F (5′-TCTCAACCAATCAGCACACC-3′), R (5′-CCTCCAAGAGTGGACA-GAGC-3′), Cyc: F (5′-CAAATGCTGGACCAAACACAA-3′), R (5′-CCATCCAGCCATTCAGTCTTG-3′).

## Immunoglobulin sequencing and analysis

Mice were treated with TPA, which was applied to the dorsal side of the ear skin twice a week for 2 weeks. 48 hr after the last exposure, draining-LNs were collected and plasma cells (FSC$^{hi}$CD95$^{hi}$CD138$^+$) cell sorted on a BD FACS Aria III (BD Biosciences, NJ, USA). RNA was extracted using RNEasy Micro kits (Qiagen) and reverse transcribed using SuperScript III Reverse Transcriptase (ThermoFisher). RNA was also collected from whole spleens of naïve mice and reverse transcribed similarly. Sequences were amplified using Phusion High-Fidelity DNA Polymerase (New-England Biolabs) to ensure accuracy and robust performance for large PCR products. PCR with a primer in the constant Cε region 5′-CTAGGGTCATGGAAGCAGTGCC-3′ in combination with a pro-miscuous V region primer 5′-GAGGTGCAGCTGCAGGAGTCTGG-3′ was performed. Primers were labelled at either end with Multiplex Identifiers (MIDs) for multiplexing. PCR thermal cycling was as follows: 98℃, 30 s; 30x (98℃, 30 s; 60℃, 30 s; 72℃, 35 s); and 72℃, 10 min. Amplicons were purified by gel extraction using QIAquick PCR purification kit (Qiagen) and deep-sequenced by long-read 454 pyrosequencing on the Genome Sequencer FLX system (Roche). Raw sequencing data were presented in FASTA format and unproductive sequences were removed (data clean-up stages were performed as described by *Wu et al., 2015*).

Analysis was performed as previously described (*Wu et al., 2015*). In short, sequences were assigned to individual samples according to their MID. V(D)J gene assignments of individual sequences were annotated using IMGT/HighV-Quest and the mouse database. Physiochemical properties

of the CDRH3 region were calculated using the R package peptides. Physiochemical properties included length, isoelectric point (pI) and frequencies of amino acid classes in the CDRH3 region.

## Bone marrow-derived basophils and mast cells

Bone marrow was obtained from tibias and femurs of 8–10-week-old BALB/c mice and erythrocytes depleted using Lysis Buffer Hybri-MaxTM (Sigma). Cells were washed and seeded in T25 flasks (TPP Tissue Culture Flasks, Sigma) in 10 ml RPMI supplemented with 10% heat inactivated foetal bovine serum (Gibco, Life Technologies), 1x penicillin streptomycin glutamine (PSG) (Gibco), 0.1 mM 2-mercaptoethanol (Gibco) and 10 ng/mL rIL-3 (PeproTech, London, UK). Cells were incubated at 37°C, 5% $CO_2$ in a humidified incubator and the media were refreshed every 5 days. Basophil numbers would peak after ~14 days and mast cells after ~1 month. For some experiments, cells were loaded with purified mouse IgE antibodies (BD) at 0.5 µg/mL for 24 hr and subsequently crosslinked with anti-IgE (R35-72, BD) at 5 µg/mL. IgE-crosslinked cells were harvested for qRT-PCR analyses and the supernatants stored.

## Primary neonatal keratinocyte cultures

Total body wall skin from neonatal mice (<5 days old) was incubated overnight at 4°C in 5 U/ml Dispase (BD) supplemented with 1x antibiotic and antimycotic solution (Sigma). The epidermis was isolated and further digested in TrypLE Express supplemented with 200 µg/ml DNAse I and DNAse buffer. Cell suspensions were filtered and resuspended in defined KC serum-free medium with supplements (Life) and 1x antibiotic–antimycotic solution. Keratinocytes were seeded at an appropriate cell density onto tissue culture vessels coated with rat-tail-derived collagen I (Sigma). Culture vessels were washed with PBS 24 hr following seeding to remove unattached cells, and provided with fresh medium with or without the addition of histamine (Sigma) or conditioned media from IgE-crosslinked basophils. In some experiments, histamine receptors were blocked using 10 µM of the $H_1R$ antagonist cetirizine dihydrochloride or the $H_4R$ antagonist JNJ 7777120 (both from Tocris).

## Cellular recruitment to the skin

To block cell recruitment to the skin, intradermal injections of 20 ng anti-TSLP, 20 ng anti-IL-3, both or isotype controls were injected into the dorsal ear pinnae in a volume of 10 µl. In other experiments, mice received i.v. injections of either 0.4 ug PTX or 0.8 ug PTX (Tocris), 50 ng anti-CXCR2 (R and D), 50 ng anti-CXCR4 (R and D) or suitable controls. Skin inflammation was initiated using a single dose of 2.5 nM of TPA to the dorsal ear skin and the skin was analysed 24 hr later.

## Chemotaxis

Bone-marrow-derived basophils were FACS-sorted as $CD45^+cKit–Fc\epsilon RI^+$ cells at day 14 of culture and $2 \times 10^5$ basophils placed in transwells with a pore size of 5 µm. 100 ng CXCL12 (Peprotech), 100 ng IL-3 (Peprotech) + 1 µg TSLP (R and D), C5a (as positive control) or media alone (negative control) was placed in the bottom chamber and numbers of basophils migrating through the transwell at 37°C was enumerated after 4 hr.

## Statistical evaluation

The statistical significance of difference between experimental groups was determined using two-tailed Student's t-test for unpaired data, one-way ANOVA multiple comparison, Log-rank Mantel-Cox test or linear regression, where appropriate, with results deemed significant at p<0.05. Stars of significance correlate to: *p<0.05; **p<0.01; ***p<0.001 and ****p<0.0001. Statistics were performed with GraphPad Prism 6.00 for Mac (GraphPad; La Jolla, CA, USA).

## Acknowledgements

We thank HR Rodewald for providing *Cpa3*[Cre/+] mice. We are indebted to the staff of the Imperial Central Biomedical Services for the care of the animals and the LMS/NIHR Imperial Biomedical Research Centre Flow Cytometry Facility for FACS support. We are grateful to M Botto and A Mowat for critical reading of the manuscript, and for the informed advice of many close colleagues. This work was supported by the Wellcome Trust (100999/Z/13/Z).

## Additional information

### Funding

| Funder | Grant reference number | Author |
|--------|------------------------|--------|
| Wellcome | 100999/Z/13/Z | Jessica Strid |

The funders had no role in study design, data collection and interpretation, or the decision to submit the work for publication.

### Author contributions

Mark David Hayes, Formal analysis, Validation, Investigation, Visualization, Methodology, Project administration; Sophie Ward, Greg Crawford, Rocio Castro Seoane, Validation, Investigation, Methodology; William David Jackson, Investigation, Methodology; David Kipling, Formal analysis, Methodology; David Voehringer, Resources; Deborah Dunn-Walters, Software, Formal analysis, Methodology; Jessica Strid, Conceptualization, Data curation, Formal analysis, Supervision, Funding acquisition, Investigation, Visualization, Methodology, Project administration

### Author ORCIDs

Jessica Strid (iD) https://orcid.org/0000-0003-3690-2201

### Ethics

Animal experimentation: All studies were approved by Imperial College AWERB (Animal Welfare and Ethical Review Body) and by the UK Home Office. Experiments involving cancer studies strictly adhered to the guidelines set out by the National Cancer Research Institute (NCRI) and Workman et al. in 'Guidelines for the Welfare and Use of Animals in Cancer Research'. All studies using animals were conducted following the Animal Research: Reporting In Vivo Experiments (ARRIVE) guidelines and were designed in keeping with the 3Rs (replacement, reduction and refinement) principles.

### Decision letter and Author response

Decision letter https://doi.org/10.7554/eLife.51862.sa1
Author response https://doi.org/10.7554/eLife.51862.sa2

## Additional files

### Supplementary files

• Transparent reporting form

### Data availability

RNA sequencing data supporting Figure 2 is available from the public repository on the National Center for Biotechnology Information's Sequence Read Archive in raw format (BioProject: PRJNA417372; BioSample accession: SAMN07985450, SAMN07985451, SAMN07985452, SAMN07985453, SAMN07985454, SAMN07985455).

The following dataset was generated:

| Author(s) | Year | Dataset title | Dataset URL | Database and Identifier |
|-----------|------|---------------|-------------|-------------------------|
| Mark David Hayes, Sophie Ward, Greg Crawford, Rocio Castro Seoane, William David Jackson, David Kipling, David Voehringer, Deborah Dunn-Walters, Jessica Strid | 2019 | Epithelial damage and tissue gd T cells promote a unique tumour protective IgE response | https://www.ncbi.nlm.nih.gov/bioproject/PRJNA417372 | NCBI BioProject, PRJNA417372 |

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
