## [Decision Letter]

**Acceptance summary:**

This is a very interesting manuscript describing how inflammation drives an increase in IgE locally and systemically together with an influx of basophils into the tissue. The authors nicely show that IgE exacerbates skin inflammation and induces histamine-mediated promotion of epithelial cell proliferation and differentiation, which is primarily mediated through FcεRI signalling in basophils. The authors also show that basophils are recruited to the inflamed skin through the action of CXCL12-CXCR4 and contributes to tumor growth.

**Decision letter after peer review:**

Thank you for submitting your article "Natural IgE promotes epithelial hyperplasia and inflammation-driven tumour growth" for consideration by *eLife*. Your article has been reviewed by three peer reviewers, including Nicola L Harris as the Reviewing Editor and Reviewer #1, and the evaluation has been overseen by Satyajit Rath as the Senior Editor. The following individual involved in review of your submission has agreed to reveal their identity: Marcus Robinson (Reviewer #3).

The reviewers have discussed the reviews with one another and the Reviewing Editor has drafted this decision to help you prepare a revised submission.

Summary:

This manuscript describes how administration of the inflammatory agent TPA drives an increase in natural IgE locally and systemically together with an influx of basophils into the tissue. The authors show that IgE exacerbates skin inflammation and induces histamine-mediated promotion of epithelial cell proliferation and differentiation, which is primarily mediated through FceRI signalling in basophils. The authors also show that basophils are recruited to the inflamed skin through the action of CXCL12-CXCR4 whereby CXCR4 is upregulated on basophils by TSLP and IL-3, the latter produced by T cells. The authors then investigated whether IgE was involved in an inflammation-driven model of cutaneous carcinogenesis. In this model, low level mutations are initiated by DMBA and tumour growth is promoted by regular exposure to TPA. They show that both IgE and FceRI are required for tumour development and suggest that it is the FceRI signalling in basophils that promotes the growth of tumour cells.

Essential revisions:

1) The authors conclude that IgE with "natural specificities" promotes tumour growth in their model (DMBA followed by repeated TPA application). This is in contrast to the autoreactive IgE-mediated protective role that was previously published by the same group using a similar cSCC model (2+ applications of DMBA) (Crawford et al., 2018.

All three reviewers agreed that although the authors clearly show that TPA increases natural IgE locally and systemically, there was insufficient evidence that this natural IgE drove the tumor growth as it was measured after only 2 weeks. By contrast, TPA was applied twice weekly for >15 weeks in the tumour model. The authors should address this question by providing evidence that the IgE repertoire does not change following the extended TPA application and following DMBA + TPA used in induce tumor growth. Should this evidence not be forthcoming or possible to generate the title and claims should be revised appropriately.

2) The authors conclude that basophils mediate the IgE-dependent tumor growth. However it remains entirely possible, and even likely based on their data, that basophils and Mast cell both contribute to tumor growth as depleting both cell types (*Cpa3^Cre^*^/+^ mic), depleting IgE signalling (*FcεRIa^-/-^*) or IgE production (*Igh7^-/-^*) was substantially more protective than depletion of basophils alone. Thus it appears that an element of redundancy may exist between mast cells and basophils, Perhaps depleting one population gives rise to more of the other at the tumour site? If the authors have data to this effect it would be interesting to include this, otherwise the possible contribution of mast cells should be discussed in the text.

3) As the recruitment via CXCR4 is a major point of the paper, it would be prudent to demonstrate basophil chemotaxis toward CXCL12 in culture--CXCR4 is important for chemotaxis using in vitro assays of basophils. This point would greatly strengthen the paper, however if such assays cannot be completed within 2 months it would be sufficient to state clearly in the text that but cells can express chemokine receptors without exhibiting chemotaxis toward its ligands (e.g. long-lived plasma cells express CXCR4 but lose the capacity to migrate along CXCL12 axes, making them unable to access the same niches as new PC).

---

## [Author Response]

Essential revisions:1) The authors conclude that IgE with "natural specificities" promotes tumour growth in their model (DMBA followed by repeated TPA application). This is in contrast to the autoreactive IgE-mediated protective role that was previously published by the same group using a similar cSCC model (2+ applications of DMBA) (Crawford et al., 2018.

It is worth noting that we previously showed that the IgE repertoire differs substantially when induced by topical exposure to the carcinogen DMBA versus by exposure to the inflammatory agent TPA (Crawford, et al., 2018). This may indeed contribute to the differing outcomes of IgE engagement in the two models (‘DMBA only model’ where tumour growth is driven by a high mutational load versus ‘DMBA-TPA model’ where tumour growth is driven by chronic inflammation). We show here that topical TPA induces IgE with natural specificities and a similar repertoire to that in resting naïve mice.

All three reviewers agreed that although the authors clearly show that TPA increases natural IgE locally and systemically, there was insufficient evidence that this natural IgE drove the tumor growth as it was measured after only 2 weeks. By contrast, TPA was applied twice weekly for >15 weeks in the tumour model. The authors should address this question by providing evidence that the IgE repertoire does not change following the extended TPA application and following DMBA + TPA used in induce tumor growth. Should this evidence not be forthcoming or possible to generate the title and claims should be revised appropriately.

We fully acknowledge that ideally the IgE repertoire should have been analysed longitudinally and we would certainly like to extend our knowledge of the IgE repertoire to tumour bearing mice in the future. However, this initial repertoire analysis took a long time to perform as the immunoglobulin repertoire is not as well annotated in mice as it is in human and the heavy chain repertoire is highly variable between inbred strains (Collins, A.M. et al., Philos Trans R Soc Lond B Biol Sci 370 (2015)). This is one of only a few studies to show a thorough repertoire analysis of IgE in the mouse and we therefore think the data adds substantially to the literature in this field. So, although we fully recognize the value of the reviewer’s suggestion, the proposed experiment would take a lot of time (~6 months to grow tumours + sequencing and analysis = ~1year) and resources and therefore is not possible within the time limit.

The text has been revised accordingly in appropriate places (i.e. the word ‘natural’ has been removed when IgE is discussed in the context of promoting inflammation-driven tumour growth). The role of IgE in promoting epidermal hyperplasia is measured within the same time-frame as the repertoire analysis is performed, so this text has been left unaltered.

2) The authors conclude that basophils mediate the IgE-dependent tumor growth. However it remains entirely possible, and even likely based on their data, that basophils and Mast cell both contribute to tumor growth as depleting both cell types (Cpa3^Cre/+^ mic), depleting IgE signalling (FcεRIa^-/-^) or IgE production (Igh7^-/-^) was substantially more protective than depletion of basophils alone. Thus it appears that an element of redundancy may exist between mast cells and basophils, Perhaps depleting one population gives rise to more of the other at the tumour site? If the authors have data to this effect it would be interesting to include this, otherwise the possible contribution of mast cells should be discussed in the text.

Thank you for your comment. We agree, particularly in the tumour experiments there may indeed be some contribution also from mast cells and some redundancy may occur in the tissue: certainly this cannot be entirely excluded. As the reviewer mention, we routinely see a more complete phenotype in the *Fcer1a^-/-^* and *Igh7^-/-^* mice than in our *Cpa3^Cre/+^* and *Mcpt8^Cre/+^* mice. However, it is worth noting (refer also to Figure 7D and F) that although depleting one population does not give rise to more of the other, there are other complexities of these models. Notably, the *Cpa3^Cre/+^* and *Mcpt8^Cre/+^* models are not complete knockout models and that the *Cpa3^Cre/+^* mice (which lack all mature mast cells) also have a significantly reduced number of basophils in inflamed skin and in tumour perilesional skin, although normal proportions of basophils in the tumours. Equally, although the *Mcpt8^Cre/+^* mice lack all basophils in inflamed skin and tumour perilesional skin, some basophils do get through to the tumours (*Mcpt8^Cre/+^* mice have normal mast cell numbers in skin/tumours). I.e. the intermediate effect on tumour susceptibility that we observe could also be due to incomplete Cre-toxicity in these models and in the case of *Cpa3^Cre/+^* mice, that the basophils are also affected. In the shorter-term skin inflammation models, *Mcpt8^Cre/+^* completely lack basophils and have normal mast cells numbers in the skin – so this data is more straight forward to interpret. A possible contribution of mast cells is discussed in the Discussion.

3) As the recruitment via CXCR4 is a major point of the paper, it would be prudent to demonstrate basophil chemotaxis toward CXCL12 in culture--CXCR4 is important for chemotaxis using in vitro assays of basophils. This point would greatly strengthen the paper, however if such assays cannot be completed within 2 months it would be sufficient to state clearly in the text that but cells can express chemokine receptors without exhibiting chemotaxis toward its ligands (e.g. long-lived plasma cells express CXCR4 but lose the capacity to migrate along CXCL12 axes, making them unable to access the same niches as new PC).

We have included additional data to strengthen this point. We have performed in vitro chemotaxis experiments toward CXCL12 using bone marrow-derived basophils (that express high levels of CXCR4). We can demonstrate significant basophil chemotaxis toward CXCL12 and that IL-3/TSLP by themselves are not chemotactic (despite that blocking these cytokines in vivo reduced basophil migration, Figure 6M, N). This new data has been added to the manuscript as an additional supplementary figure (Figure 6—figure supplement 1).